# Targeting transcription in heart failure via CDK7/12/13 inhibition

Austin Hsu [1,2,13], Qiming Duan[1,13], Daniel S. Day[3,13], Xin Luo[4], Sarah McMahon[1,2], Yu Huang[1],
Zachary B. Feldman[5], Zhen Jiang[1,4], Tinghu Zhang[6,7], Yanke Liang[7,8], Michael Alexanian[1], Arun Padmanabhan[1,9],
Jonathan D. Brown[5], Charles Y. Lin [10], Nathanael S. Gray [6,7,12], Richard A. Young [3,8],
Benoit G. Bruneau [1,11,14✉] & Saptarsi M. Haldar [1,4,9,14✉]

Heart failure with reduced ejection fraction (HFrEF) is associated with high mortality, high-lighting an urgent need for new therapeutic strategies. As stress-activated cardiac signaling cascades converge on the nucleus to drive maladaptive gene programs, interdicting pathological transcription is a conceptually attractive approach for HFrEF therapy. Here, we demonstrate that CDK7/12/13 are critical regulators of transcription activation in the heart that can be pharmacologically inhibited to improve HFrEF. CDK7/12/13 inhibition using the first-in-class inhibitor THZ1 or RNAi blocks stress-induced transcription and pathologic hypertrophy in cultured rodent cardiomyocytes. THZ1 potently attenuates adverse cardiac remodeling and HFrEF pathogenesis in mice and blocks cardinal features of disease in human iPSC-derived cardiomyocytes. THZ1 suppresses Pol II enrichment at stress-transactivated cardiac genes and inhibits a specific pathologic gene program in the failing mouse heart. These data identify CDK7/12/13 as druggable regulators of cardiac gene transactivation during disease-related stress, suggesting that HFrEF features a critical dependency on transcription that can be therapeutically exploited.

[1] Gladstone Institutes, San Francisco, CA, USA. [2] Biomedical Sciences Graduate Program, San Francisco, CA, USA. [3] Whitehead Institute for Biomedical Research, Cambridge, MA, USA. [4] Amgen Research, South San Francisco, CA, USA. [5] Vanderbilt University Medical Center, Nashville, TN, USA. [6] Department of Cancer Biology, Dana-Farber Cancer Institute, Boston, MA, USA. [7] Department of Biological Chemistry & Molecular Pharmacology, Harvard Medical School, Boston, MA, USA. [8] Department of Biology, Massachusetts Institute of Technology, Cambridge, MA, USA. [9] Department of Medicine, Cardiology Division, UCSF, San Francisco, CA, USA. [10] Baylor College of Medicine, Houston, TX, USA. [11] Department of Pediatrics and Cardiovascular Research Center, UCSF, San Francisco, CA, USA. [12] Present address: Department of Chemical and Systems Biology, The Cancer Therapeutics Research Program, and Innovative Medicines Accelerator, Stanford University, Palo Alto, CA, USA. [13] These authors contributed equally: Austin Hsu, Qiming Duan, Daniel S. Day. [14] These authors jointly supervised this work: Benoit G. Bruneau, Saptarsi M. Haldar. ✉email: benoit.bruneau@gladstone.ucsf.edu; shalda01@amgen.com

Despite current standard of care, heart failure with reduced ejection fraction (HFrEF) remains associated with excessively high mortality and morbidity, highlighting an urgent need for new therapeutic strategies. Several pharmacotherapies for HFrEF that have been proven to improve outcomes, such as beta-adrenergic receptor antagonists and inhibitors of the renin-angiotensin system, typically interdict signaling at or near the plasma membrane. Ultimately, these stress-activated signaling cascades converge on the nucleus to drive maladaptive changes in gene transcription, which can chronically compromise cardiac function and fuel the vicious cycle of adverse cardiac remodeling and HFrEF pathogenesis[1]. As the gene regulatory machinery functions as a nodal integrator of upstream stress signals, drugging specific components of the transcription apparatus represents a conceptually attractive approach to treat HFrEF.

Previous studies have shown that hyperphosphorylation of RNA Polymerase II (Pol II) by multiprotein kinase complexes such as PTEFb (positive transcription elongation factor b) is a hallmark of pathological cardiac stress in rodent models of heart failure and in failing human hearts[2,3], prompting us to hypothesize that inhibition of additional upstream Pol II C-terminal domain (CTD) kinase complexes such as TFIIH could attenuate pathologic transcription during HFrEF pathogenesis. THZ1 is a first-in-class small molecule inhibitor of CDK7 (Cyclin dependent kinase 7), a core kinase in the eukaryotic TFIIH complex upstream of Pol II CTD phosphorylation. THZ1 covalently binds Cysteine-312 on CDK7 at a site spatially distinct from the catalytic pocket, resulting in potent allosteric inhibition of CDK7 kinase activity[4]. THZ1 also inhibits two related Pol II CTD kinases, CDK12 and CDK13, by covalently binding accessible cysteine residues that are homologous to C312 of CDK7[4]. Preclinical studies across a broad array of tumor types have demonstrated that THZ1 can potently suppress transcription of gene programs that drive cancer progression[4–10]. These studies prompted us to explore the role of CDK7/12/13-dependent transcription in HFrEF pathogenesis. Here, we demonstrate that CDK7/12/13 are critical regulators of stress-mediated transcription activation in cultured cardiomyocytes and the adult mouse heart that can be pharmacologically inhibited to improve HFrEF.

## Results

### CDK7/12/13 inhibition in cultured cardiomyocytes inhibits the hypertrophic stress response.

We began by studying the effects of inhibiting CDK7/12/13-dependent Pol II CTD phosphorylation via THZ1 (structure shown in Fig. 1A) in cultured neonatal rat ventricular myocytes (NRVM), a well-established in vitro system to probe adverse cardiomyocyte remodeling[11]. THZ1 treatment potently attenuated α1-adrenergic agonist (phenylephrine, PE)-induced cellular hypertrophy (Fig. 1B) and induction of Nppa and Nppb, two hallmark markers of pathological cardiomyocyte stress (Fig. 1C). To determine a specific role for cardiomyocyte CDK7/12/13 activity on phosphorylation of specific serine residues in the C-terminal heptapeptide repeats of Pol II[12,13], we performed Western blots using phospho-serine specific Pol II antibodies. We observed increased levels of Ser2P, Ser7P and Ser5P Pol II phosphoforms during PE-mediated hypertrophic stress (Fig. 1D), consistent with increased Pol II CTD phosphorylation in this setting. THZ1 attenuated the PE-dependent increase in Ser2P and Ser7P (Fig. 1D). We did not observe a significant effect on bulk abundance of Ser5P in cardiomyocytes, which may reflect redundant regulation of this phosphoform by other Pol II CTD kinases[14–17]. The $IC_{50}$ of THZ1 in these cardiomyocyte assays was 5–10 nM, which reflects a higher sensitivity to THZ1 than what has been observed in studies of growth inhibition of several cancer cell types[5–9,18–26] (Supplementary Fig. 1A). Under baseline conditions, THZ1 showed no significant effect on NRVM size, Nppa/Nppb

expression, or cell death (Fig. 1B–D and Supplementary Fig. 1B). As a critical negative control, we used THZ1-R, an inactive structural analog of THZ1 with minimal kinase inhibitory activity[4]. THZ1-R did not affect cellular hypertrophy, Nppa/Nppb induction, or Pol II CTD phosphorylation (Supplementary Fig. 1C–E). To validate for our findings using THZ1, we also assessed the anti-hypertrophic effect of the small molecule probe YKL-1-116[27] (Supplementary Fig. 2A), which has substantially lower potency against CDK12/13. YKL-1-116 inhibited cellular hypertrophy and Nppa/Nppb induction but required concentrations 50–100 fold higher than THZ1 (Supplementary Fig. 2B, C)[27]. These data suggest that potent and combined inhibition of CDK7/12/13 is required for maximal suppression of the hypertrophic stress response in cardiomyocytes.

To provide orthogonal validation for the on-target molecular pharmacology of THZ1 in our experimental system, we utilized specific siRNA probes to individually knockdown Cdk7, Cdk12, or Cdk13 as well as triple knockdown (TKD) of all three kinases (Supplementary Fig. 3). We found that individual knockdown of Cdk7, 12, or 13 did not consistently attenuate cardiomyocyte hypertrophy, Nppa and Nppb induction, and Pol II CTD hyperphosphorylation with the same potency of THZ1, indicating that depletion of a single kinase did not fully recapitulate the robust effects of THZ1 (Supplementary Fig. 3E, F). In contrast, triple knockdown of Cdk7/12/13 potently attenuated PE-induced hypertrophy, Nppa/Nppb expression, and Pol II CTD hyperphosphorylation in a manner that more closely approximated the effects of THZ1 (Fig. 1E–H). The increased abundance of these specific Pol II phosphoforms during agonist stimulation and its decrease with CDK7/12/13 inhibition using THZ1 or siRNA confirms that the combined activity of these kinases is specifically increased during hypertrophic stress in cardiomyocytes. Together, our complementary data using chemical probes and RNAi establish a role for CDK7/12/13 as druggable transcription co-activators and positive regulators of pathological cardiomyocyte hypertrophy.

To enhance relevance to human HFrEF, we ascertained whether CDK7/12/13 inhibition with THZ1 could also block pathologic remodeling in human cardiomyocytes. We tested the effects of THZ1 in human-induced pluripotent stem cell-derived cardiomyocytes (iPSC-CMs), a well-validated in vitro experimental platform to study human cardiomyocyte stress signaling, using an iPSC line derived from a healthy human donor [iCell Cardiomyocytes, Cellular Dynamics International Inc. (CDI)][28]. Previous studies have demonstrated that these cells mount a hypertrophic response to 10 nM endothelin-1 (ET-1)[28,29]. We found that THZ1 attenuated ET-1–mediated hypertrophic growth in human iPSC-CMs in a dose-dependent manner (Fig. 1I, J) with an $IC_{50}$ ~10 nM, a potency that paralleled our rodent cardiomyocyte data. Consistent with the effects on cellular hypertrophy, qRT-PCR demonstrated that THZ1-suppressed transactivation of NPPB/BNP (Fig. 1K) and other typical marker genes induced in human iPSC-CMs during ET-1 stimulation (Supplementary Fig. 4A)[28]. Quantitative enzyme-linked immunosorbent assay (ELISA) confirmed that THZ1 attenuated the ET-1–stimulated secretion of N-terminal pro-BNP protein, a widely used clinical biomarker for human HFrEF (Fig. 1L).

### THZ1 alters gene expression and transcription in NRVM during hypertrophic stress.

We next performed RNA-Seq in NRVM to profile the genome wide transcriptional mechanisms underlying the protective effects of THZ1 (Fig. 2A, Supplementary Data 1). Hierarchical clustering revealed six distinct gene expression patterns, with approximately one-fourth of all expressed

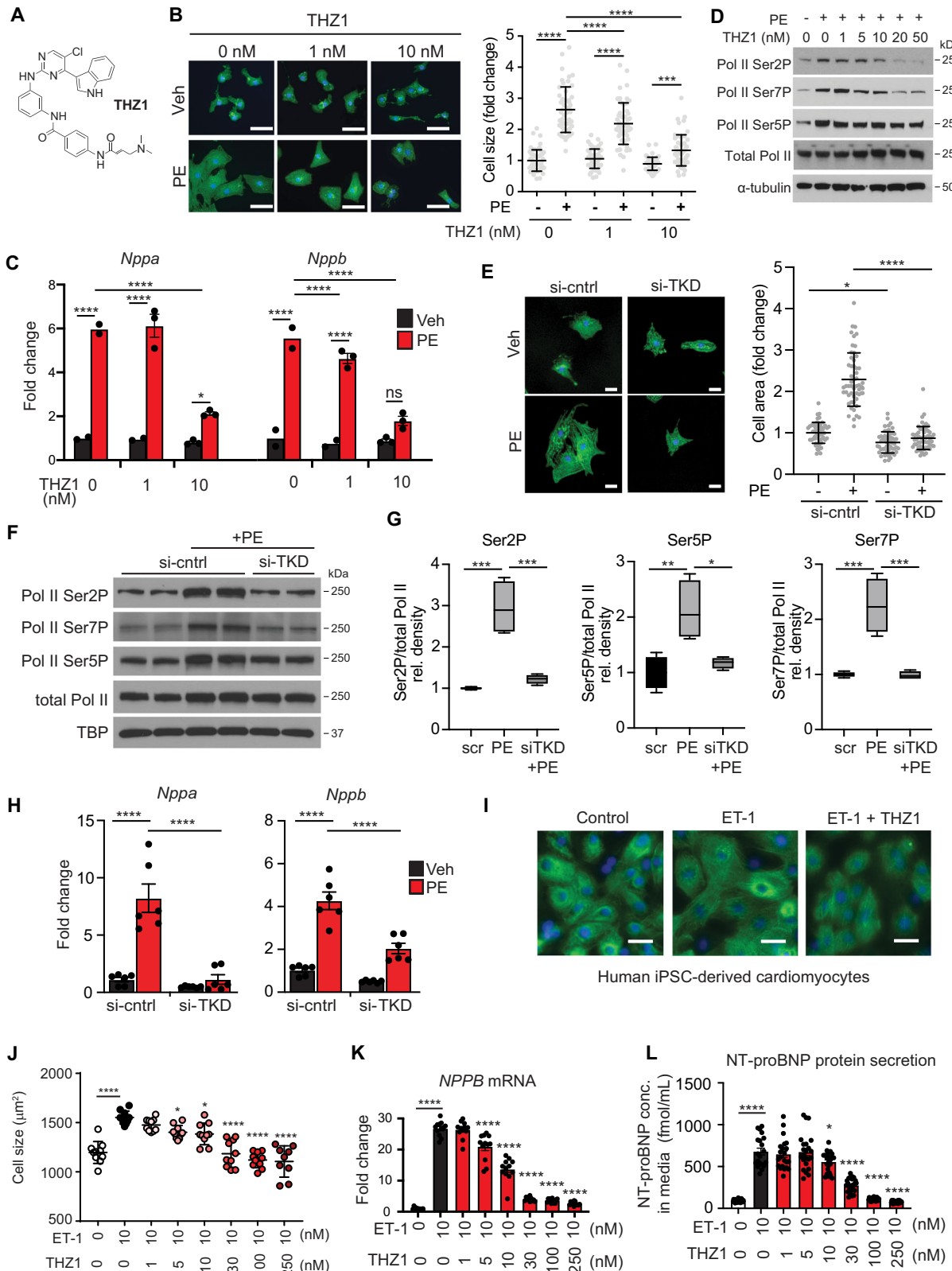

genes falling into cluster 5, representing genes induced by PE stimulation and robustly suppressed with THZ1 co-treatment, such as *Nppa*, *Nppb*, and *Xirp2/Myomaxin* (Fig. 2B). THZ1 also attenuated the PE-mediated downregulation of a subset of genes (cluster 2), further suggesting that primary inhibition of stress-gene transactivation can shift the transcriptome towards the basal state. While THZ1 had no major effect on cardiomyocyte growth

or viability at baseline (Fig. 1B–D and Supplementary Fig. 1B), there was a subset of genes for which THZ1 altered baseline expression (clusters 1 and 3). Gene ontology analysis for each cluster is provided in Supplementary Data 2. Given the role of CDK7/12/13 as activators of Pol II and the robust ability of THZ1 to suppress stress-induced cardiomyocyte hypertrophy, we focused our attention on PE-mediated gene transcription by

**Fig. 1 CDK7/12/13 inhibition attenuates hallmark features of pathologic cardiomyocyte remodeling in vitro. A** Chemical structure of THZ1.
**B** Representative image of NRVM treated ± THZ1 (at indicated concentrations) and PE (100 μM) for 48 h with cell area quantification ($n = 50$–55 randomly selected cells per condition). Scale bar: 30 μm. Bars denote mean ± SD. ***$p = .0002$. **C** Quantification of expression levels of indicated genes from real-time PCR ($n = 3$ for THZ1 1 nM + PE, and THZ1 10 nM ± PE groups; $n = 2$ for THZ1 0 nM ± PE and THZ1 1 nM − PE). Experiment was repeated three independent times with similar results. **D** Representative Western blots of NRVM with indicated treatment for specific targets. $N = 3$ independent experiments. Pol II phosphoforms and total Pol II were each probed from a separately run gel and membrane, with each lane loaded with an equivalent amount of total protein and lysate volume, derived from a common master stock of protein lysate for each designated condition. The loading control (α-tubulin) was probed on the same blot with Pol II Ser7P. For these Western blots, NRVM were harvested 30 min after exposure to PE. **E** Representative images of NRVM treated ± si-control (si-cntrl) or si-*Cdk7/12/13* (TKD) and PE (100 μM) for 48 h with cell area quantification ($n = 55$ randomly selected cells per condition). Scale bar: 30 μm. Bars denote mean ± SD. *$p = 0.0127$. **F** Representative Western blots of NRVM with indicated treatment for specific targets. Pol II phosphoforms and total Pol II were each probed from a separately run gel and membrane, with each lane loaded with an equivalent amount of total protein and lysate volume, derived from a common master stock of protein lysate for each designated condition. The loading control (TBP) was probed on the same blot with Pol II Ser7P. For these Western blots, NRVM were harvested 30 min after exposure to PE. Experiment was repeated three independent times with similar results. **G** Densitometry of Western blots in Fig. 1F ($n = 4$ per condition). Box plots show center line as median, whiskers show maxima and minima, and box limits show upper and lower quartiles. ***$p = 0.0002$ for Ser2P: scr vs. PE, ***$p = 0.0004$ for Ser2P: PE vs. siTKD + PE, **$p = 0.0048$ for Ser5P: scr vs. PE, *$p = 0.0131$ Ser5P: PE vs siTKD + PE, ***$p = 0.0006$ for Ser7P: scr vs PE, ***$p = 0.0005$ for Ser7P: PE vs siTKD + PE. **H** Quantification of expression levels of indicated genes from real-time PCR ($n = 4$). **I, J** Representative hiPSC-CM images (alpha-actinin staining in green and DAPI in blue) and quantification of cell size for indicated treatments ($n = 10$ for all conditions except ET-THZ1–250nM for which $n = 9$). Each data point represents the mean cell area for a single cell-culture chamber. For each chamber, 25 cells were randomly selected for area quantification. Scale bar: 30 μm. Bars denote mean ± SD. Significance values represent comparisons against the group that was exposed only to ET-1 (10 nM). *$p = 0.0395$ for ET vs. ET-THZ1–5nM, *$p = 0.0133$ for ET vs. ET-THZ1–10nM. **K** Gene expression levels of *NPPB* in hiPSC-CM with indicated treatments ($n = 10$). Significance values represent comparisons against the group that was exposed only to ET-1 (10 nM). **L** Protein concentration of secreted NT-proBNP in hiPSC-CM in the culture medium ($n = 10$). Significance values represent comparisons against the group that was exposed only to ET-1 (10 nM). *$p = 0.0281$ for ET vs ET-THZ1–10nM. Data are shown as means ± SEM unless otherwise noted. *$p < 0.05$, **$p < 0.01$, ***$p < 0.001$, ****$p < 0.0001$ for all indicated comparisons. One-way ANOVA with Tukey's multiple comparisons test was used for all statistical analyses. Exact p values are noted when possible. Source data are provided as a Source Data file.

performing ChIP-Seq for total Pol II in NRVM after 48 h of PE stimulation. Representative gene tracks for the *Nppa/Nppb* locus (Fig. 2C) showed that PE stimulation increased gene body enrichment of Pol II, consistent with stress-activated transcription of these genes. THZ1 attenuated enrichment of Pol II at the *Nppa* and *Nppb* loci, consistent with inhibition of gene transcription. Genome-scale quantification of Pol II density at this 48-h time point revealed that THZ1 led to a modest increase in Pol II accumulation at transcription start sites (Fig. 2D), a finding that suggests a net effect of impaired transcription elongation at this late time point. We assessed the Pol II traveling ratio (defined here as the ratio of Pol II occupancy at the TSS over the gene body; Fig. 2E), a widely used indicator of genome-scale transcription elongation[30,31], for genes regulated by THZ1 during PE-mediated stress. THZ1 elicited a rightward shift in the Pol II traveling ratio curve, also suggesting a net effect of THZ1 in attenuating Pol II elongation in pathologically stressed cardiomyocytes after 48 h of PE stimulation (Fig. 2E). These ChIP-Seq data are consistent with the biochemical data demonstrating CDK7/12/13 inhibition blocks stress-dependent accumulation of bulk Pol II Ser2P abundance (Fig. 1D, F), a post-translational modification indicative of elongating Pol II. We note that Pol II ChIP-seq in primary cultured neonatal cardiomyocytes, which was performed under experimental conditions that are required for robust agonist-induced hypertrophic responses (low plating density and prolonged serum starvation), did not provide signal robustness that would allow us to make reliable correlations between locus-specific Pol II enrichment and differential gene expression by RNA-seq. Furthermore, we cannot exclude a primary and rapid effect of THZ1 on Pol II initiation, which can influence subsequent rates of Pol II elongation and is an event in the transcription cycle that likely occurs at much earlier timepoints after agonist stimulation compared to the 48-h time point used in this experiment. Together, these in vitro data support that CDK7/12/13 are cell-autonomous effectors of transcription activation during cardiomyocyte stress and that CDK7/12/13 inhibition attenuates hallmark features of pathologic cardiomyocyte remodeling.

**THZ1 improves cardiac function and inhibits a maladaptive gene expression program in a mouse model of heart failure.** Our findings in cultured cardiomyocytes prompted us to test whether interdicting CDK7/12/13-dependent transcription using THZ1 could block HFrEF pathogenesis in vivo. We subjected mice to transverse aortic constriction (TAC), a widely used surgical model of left ventricle (LV) pressure overload and progressive HFrEF[32], with administration of THZ1 (20 mg/kg/day, intraperitoneally) or vehicle for 60 days (Fig. 3A). This THZ1 dose has been validated to provide target coverage and efficacy in murine tumor xenograft studies and is generally well tolerated in mice[4,6,25], allowing us to test for proof-of-concept in vivo in the context of HFrEF, with the caveat that detailed toxicological analysis has not yet been reported for this chemical probe. We found that THZ1 attenuated several hallmark features of adverse cardiac remodeling and HFrEF pathogenesis including LV systolic dysfunction (Fig. 3B, C and Supplementary Fig. 4B), cardiomegaly (Fig. 3D, E), LV wall thickening (Supplementary Fig. 4C), LV cavity dilation (Supplementary Fig. 4D), cardiomyocyte hypertrophy (Fig. 3F), and LV fibrosis (Fig. 3G). THZ1 had no significant effect on cardiac mass, cardiac structure or LV function in the sham group (Fig. 3E, Supplementary Fig. 4 B–D). In addition, THZ1 had no effect on body weight (Supplementary Fig. 4E), consistent with the general tolerability of this compound that has been observed in mouse cancer xenograft studies. THZ1 did not affect systolic or diastolic blood pressure in mice (Supplementary Fig. 4E–G), suggesting that its salutary effect in HFrEF pathogenesis was not simply due to modulation of systemic blood pressure. To confirm on-target bioactivity in vivo, we performed Western blotting for Pol II CTD phosphorylation from LV tissue samples. TAC led to increased abundance of Pol II Ser2P and Ser7P, which were both attenuated by THZ1 (Supplementary Fig. 4H), supporting that cardiac CDK7/12/13 activity is increased during stress and blocked by THZ1 in vivo.

Next, we performed RNA-Seq from LV tissue in each experimental group (Fig. 4A). PCA clustering showed an overlap of Sham-Veh and Sham-THZ1 groups (Fig. 4B), revealing that

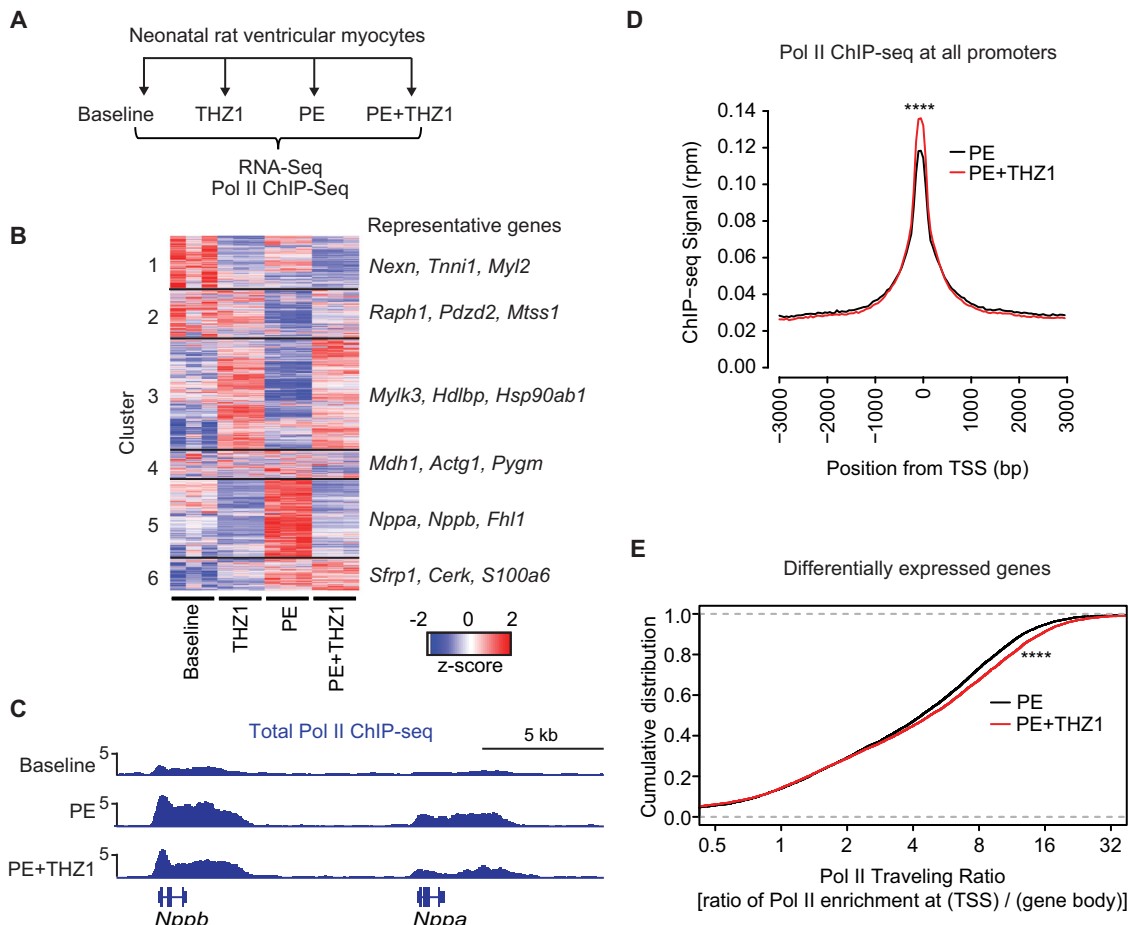

**Fig. 2 THZ1 blocks stress-responsive gene programs and Pol II enrichment in cultured cardiomyocytes. A** Schema of the experimental design. ChIP-seq for total Pol II was performed for 3 conditions: baseline, PE, and PE + THZ1. **B** A heatmap showing clustered, row-normalized global gene expression profile in each sample in the indicated treatment group (*n* = 3) with representative gene callouts. **C** Representative gene tracks of Pol II enrichment for canonical hypertrophic marker genes (*Nppb* and *Nppa*). **D** Accumulative Pol II occupancy on all promoters. **E** Empirical cumulative distribution plots of Pol II traveling ratios (TR) for genes that are differentially expressed in response to PE stimulation (PE vs baseline). TR is defined as the ratio of (Pol II signal at TSS) / (Pol II signal at gene body). ****p < 0.0001 for indicated comparisons. Source data are provided as a Source Data file.

intermittent THZ1 exposure at the doses used here had minimal effect on the transcriptome of non-diseased hearts. This finding is consistent with the observation that THZ1 treatment did not alter cardiac mass, cardiac structure, or LV systolic function in the sham group (Fig. 3). TAC-mediated stress shifted the cardiac transcriptome along PC1, and the major effect of THZ1 treatment during TAC was to revert the gene expression profile back towards the sham group along PC1 (Fig. 4B). These shifts along PC1 encompassed a broad program of stress-induced genes whose upregulation was blunted by THZ1 (Fig. 4C), including canonical markers of human HFrEF such as *Nppa*, *Nppb*, *Ctgf*, and *Rcan1* (Fig. 4D). We identified a set of 800 genes that were upregulated during TAC, of which 401 genes (50%) were significantly suppressed by THZ1 (Fig. 4E and Supplementary Data 3). This gene set showed strong enrichment for functional terms representing several pathological processes that are hallmarks of human HFrEF (Supplementary Fig. 5A)[33], including cellular growth/anabolism, matrix remodeling, inflammation/fibrosis, and reactivation of striated muscle developmental programs. These findings were corroborated by gene set enrichment analyses, which showed that the THZ1-suppressed gene program was strongly enriched for targets of EGFR, IL6 and TGF-β signaling (Supplementary Fig. 5B). Together, these data demonstrate that THZ1 can blunt stress-dependent transcription in the heart and suppress hallmark features of pathologic cardiac remodeling and HFrEF progression in vivo.

**Systemic THZ1 administration affects gene expression in multiple cell compartments in the failing mouse heart**. While our in vitro data in primary rodent and human iPSC-derived cardiomyocytes support a cardiomyocyte-intrinsic effect of CDK7/12/13 inhibition, we recognize that administration of THZ1 in vivo may simultaneously act on multiple cell compartments that populate the stressed heart, including cardiomyocytes. Our bulk RNA-seq from adult mouse LV tissue suggests that THZ1 affects genes expressed in both cardiomyocytes and non-cardiomyocyte compartments during TAC. To better understand the relative responsiveness of various myocardial cellular compartments to THZ1 treatment during TAC, we curated two published single-cell transcriptomic datasets from adult mouse hearts[34,35] to generate a cellular atlas (Fig. 4F, G) that allows for compartment-specific "fingerprinting" of our bulk RNA-seq data from LV tissue. Using the set of 401 genes identified to be upregulated in TAC and suppressed with THZ1 treatment in our bulk LV tissue RNA-seq, we cross-referenced these transcripts against our clustered single cell reference dataset to extrapolate the specific cellular compartments from which these dynamically responsive transcripts originated (Fig. 4G; cell compartment-specific heatmaps shown in Supplementary Fig. 6). Consistent with our cell-culture studies, we observed a clear cardiomyocyte signature demarcated by canonical HF-associated genes (*Ankrd1*, *Xirp2*, *Nppa*, *Nppb*). We also

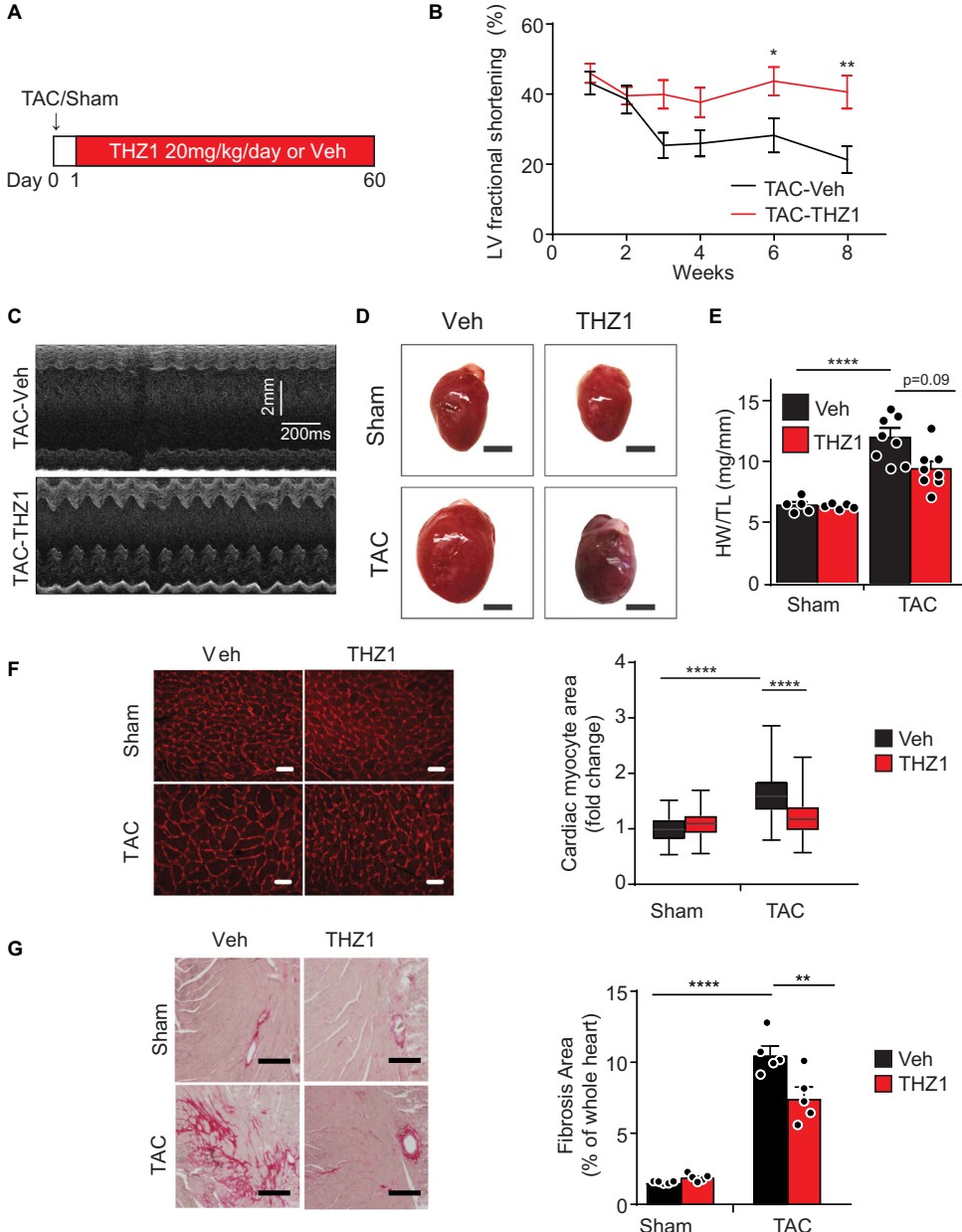

**Fig. 3 THZ1 ameliorates cardinal features of adverse cardiac remodeling and HFrEF pathogenesis in mice. A** Schema of the experimental design. **B** Two-dimensional echocardiographic quantification of left ventricle area fractional shortening in indicated groups ($n = 8$). Echocardiography for the sham groups (Sham-Veh and Sham-THZ1) were performed at the end of the study (week 8) and are plotted alongside the week 8 TAC data in Supplementary Fig. 4A–C. *$p = 0.0339$, **$p = 0.0039$. Two-way ANOVA with Holm–Sidak multiple comparisons correction was used for statistical analysis. **C** Representative M-mode echocardiographic images obtained at mid-papillary muscle level from parasternal short-axis view (8 weeks post-TAC). **D**, **E** Representative heart images and quantification of heart weight/tibial length (HW/TL) ratio at 8 weeks post-TAC. Scale bar: 3 mm. **F** Representative images of myocardium visualized by wheat germ agglutinin staining (red) and quantification of cardiomyocyte cross-sectional area in indicated groups at 8 weeks post-TAC (pooled analysis of 150 individual cardiomyocytes randomly selected from hearts of 3 independent animals per experimental condition). Scale bar: 20 μm. Data presented as standard box-and-whisker plots showing median cardiomyocyte area (horizontal line). Upper and lower quartiles are designated by the box and whiskers designate upper and lower extremes. **G** Representative heart cross sections stained with picrosirius red with quantification of fibrosis area in indicated groups ($n = 5$; 8 weeks post-TAC). **$p = 0.0027$. Scale bar: 200 μm. *$p < 0.05$, **$p < 0.01$, ***$p < 0.001$, ****$p < 0.0001$ for all indicated comparisons. Data are shown as means ± SEM unless otherwise noted. One-way ANOVA with Tukey's multiple comparisons test was used for all statistical analyses unless noted. Exact p values are noted when possible. Source data are provided as a Source Data file.

detected a robust signature of genes that were differentially expressed during TAC and sensitive to THZ1 that originated from resident cardiac fibroblasts, particularly the group of *Periostin*-positive cells that demarcate the myofibroblast sub-population. Other cell compartments in the adult mouse heart that were sensitive to THZ1 included epicardial, endothelial and endocardial cells. Taken together, these results suggest that THZ1 exerts protective effects in the TAC model by inhibiting CDK7/12/13-dependent cell state transitions across cardiomyocyte and non-cardiomyocyte cellular compartments.

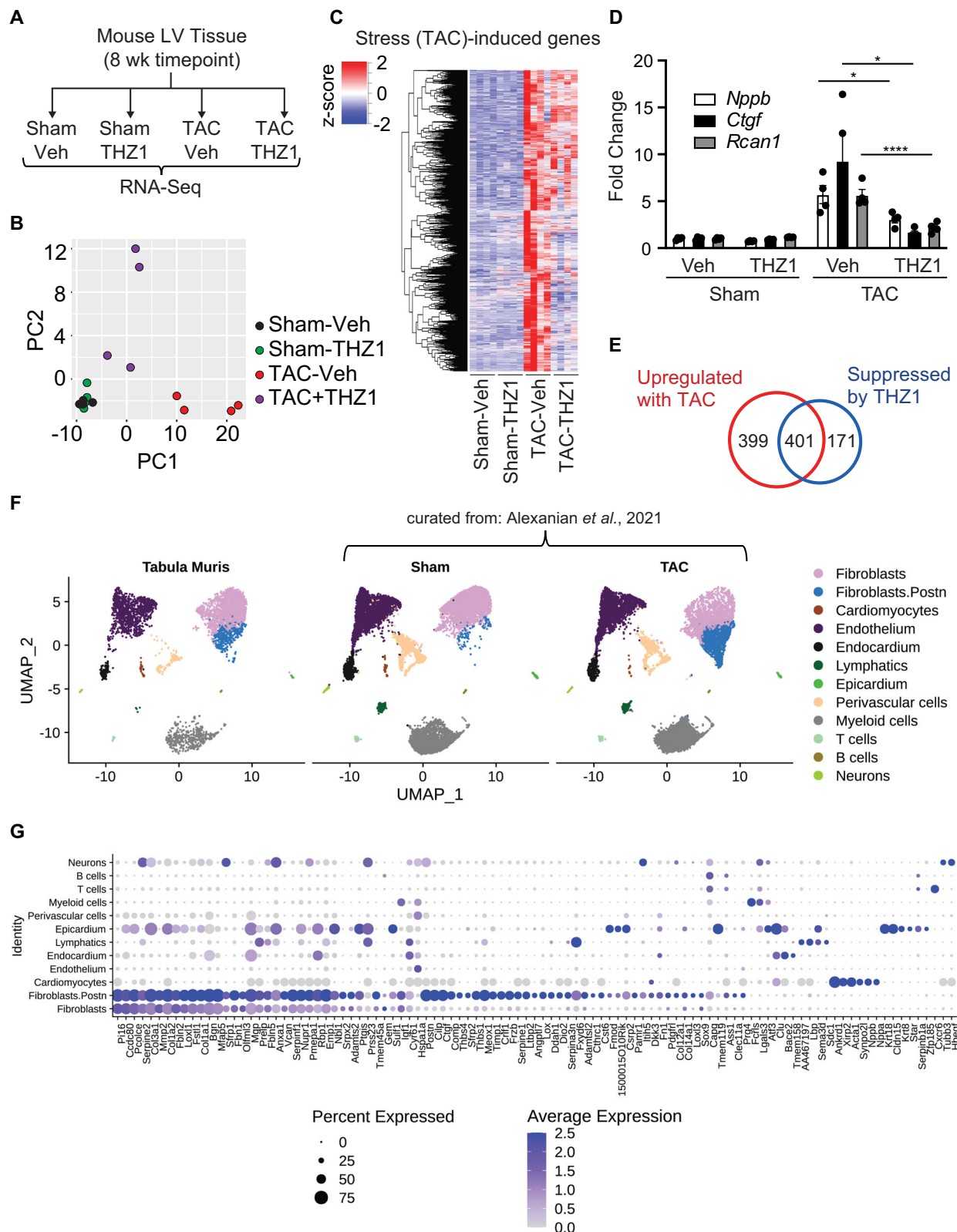

## Discussion

Collectively, the data presented here demonstrate that CDK7/12/13 activity is a critical effector of stress-dependent cardiac transcription and HFrEF pathogenesis. Using the first-in-class inhibitor THZ1, we provide proof of concept that transcription inhibition can exert effects on the heart during HFrEF pathogenesis in vitro and in vivo. Our in vitro data using chemical inhibitors and siRNA are consistent with a mechanism where CDK7, 12 and 13 have non-overlapping functions in the transcription cycle in cardiomyocytes. In this manner, multi-CDK inhibition of CDK7, 12 and 13 using THZ1 or siRNA is likely exerting additive or synergistic effects on cardiac stress responses by impairing their multiple, non-overlapping functions in transcription control, supporting the concept that inhibition of all

**Fig. 4 THZ1 ameliorates specific pathological gene programs in diverse cardiac cellular compartments in failing mouse hearts. A** Schema of the experimental design. **B** Principal component (PC) analysis showing global gene expression in indicated samples. **C** A heatmap showing clustered, row-normalized stress-induced gene expression profile in each sample in indicated group ($n = 4$). **D** Gene expression levels of indicated representative genes ($n = 4$) calculated from normalized DeSeq2 counts. Significance levels were based on the adjusted $p$-values reported by DeSeq2. Bars denote mean ± SD. *$p = 0.0171$ for Nppb: TAC-Veh vs TAC-THZ1, *$p = 0.0178$ for Ctgf: TAC-Veh vs TAC-THZ1. **E** Venn diagram demonstrating number of differentially expressed genes in each indicated compartment (cutoffs are specified in Methods section). *$p < 0.05$, ****$p < 0.0001$ for all indicated comparisons. **F** UMAP (Uniform Manifold Approximation and Projection) projection of cardiac cell compartments in mouse heart identified by integrating two public mouse heart datasets ($n = 32,407$ cells). Each dot represents a single cell. Different cell-type clusters are color-coded. **G** Gene expression distribution of cell-type enriched genes whose expression is increased with TAC and suppressed by THZ1 (in **E**) across cell compartments in adult mouse heart. Genes that were induced with TAC and suppressed by THZ1, as determined by bulk RNA sequencing, were cross-referenced with the integrated single cell dataset (in **F**) to determine the most likely cell-type compartment from which each transcript originated. The size of the dot indicates the percentage of cells with at least one compartment-defining transcript detected and the color of the dot represents the scaled average expression level of expressing cells. Representative heatmaps for differentially expressed genes emanating from each identified cell compartment are shown in Supplementary Fig 6. One-way ANOVA with Tukey's multiple comparisons test was used for all statistical analyses. Exact $p$ values are noted when possible. Source data are provided as a Source Data file.

three kinases is required to consistently and potently suppress adverse cardiac remodeling. Future studies using cardiovascular cells engineered to express a CDK7 cysteine substitution allele (Cys312Ser) that is resistant to covalent chemical inhibitors like THZ1 (or equivalent cysteine substitution alleles for CDK12/13), will also help dissect the non-overlapping functions of these three kinases in cardiac stress responses. While THZ1 can suppress agonist-induced hypertrophy in cultured cardiomyocytes, our transcriptomic profiling of adult mouse hearts demonstrates that the protective effects of THZ1 in vivo are associated with dynamic gene expression changes in both cardiomyocyte and non-cardiomyocyte compartments. Often, dissecting the causality of cell-specific contributions during heart failure pathogenesis can be interrogated using conditional gene-deletion approaches. However, modeling the pharmacology of THZ1 in vivo using conditional gene-deletion approaches will be particularly challenging because it would likely require simultaneous postnatal deletion of multiple loci (*Cdk7*, *Cdk12* and *Cdk13*). Furthermore, it is expected that gene deletion in any one cellular compartment would only give a partial protection from heart failure pathogenesis and that simultaneous CDK7/12/13 inhibition in multiple cell types, as occurs with THZ1, is required for full therapeutic effects. Finally, in contrast to the intermittent and dose-titratable effects of a chemical probe like THZ1, permanent gene deletion of these 3 kinases may not be as well tolerated. While there are certain caveats to interpreting the cell compartment and gene-specific effects of THZ1 in vivo, this study highlights some of the advantages of probing disease pathobiology using chemical biological tools.

Previous studies demonstrated that the CDK9/P-TEFb complex played an important role in cardiac hypertrophy[2]. Using pleiotropic chemical inhibitors such as **5,6-dichloro-1-β-D-ribo-furanosylbenzimidazole (DRB)** or flavopiridol or via adenoviral overexpression of a dominant-negative CDK9 construct, CDK9 inhibition was shown to attenuate cardiomyocyte hypertrophy and Pol II hyperphosphorylation in cultured neonatal rat cardiomyocytes in vitro. In addition, cardiomyocyte-specific transgenic overexpression of Cyclin-T1, a key component of the CDK9-activating complex, was sufficient to drive pathological cardiac hypertrophy in mice in vivo. These seminal findings suggested a role for the CDK9/PTEF-b transcriptional complex in cardiac hypertrophy, although the therapeutic potential of Pol II kinase inhibition could not be assessed due to lack of potent and specific chemical probes that were suitable for in vivo administration. Interestingly, adenoviral overexpression of a dominant-negative CDK7 construct was also shown to modestly attenuate agonist-induced protein synthesis in cultured cardiomyocytes[2], suggesting parallel approaches to interdict transcription during cardiomyocyte stress responses. Our study demonstrates that

simultaneous inhibition of CDK7-, 12- and 13-dependent transcription is required for potent and consistent antihypertrophic responses and establishes proof-of-concept that CDK7/12/13 inhibition using THZ1 can ameliorate pressure overload-induced heart failure in adult mice.

Our finding that THZ1 can also suppress pathologic remodeling in human iPSC-CMs at low nanomolar concentration supports the concept that manipulating stress-mediated transcriptional signaling might be an approach to limit adverse cardiac remodeling during human HFrEF pathogenesis. However, we acknowledge that systemic exposure to molecules such as THZ1 may have on-target toxicity in extracardiac organs, including effects on neuronal plasticity[36] and possibly in highly proliferative tissue compartments such as the intestinal epithelium or bone marrow. Given the generally high bar for safety required for chronically dosed cardiovascular therapeutics, future studies detailing the on-target liabilities of CDK7/12/13 inhibition and the precise cell compartments mediating therapeutic efficacy will be required to refine such a strategy in the treatment of heart failure. In contrast to several cancer drugs that cause cardiotoxicity[37], our data suggest that molecules like THZ1 may be a privileged class of anticancer therapeutics that have cardioprotective properties. More broadly, this work supports the contention that HFrEF pathogenesis, like cancer, features a general dependency on transcription that might be therapeutically exploited.

## Methods

All animal studies performed in this study were reviewed and approved by the University of California, San Francisco Institutional Animal Care and Use Committee. All studies were conducted in strict accordance with the National Institutes of Health Guide for the Care and Use of Laboratory Animals[38].

**Animal models**. Studies were conducted with age-matched male C57Bl/6J mice (The Jackson Laboratory, catalog no. 000664). Mice were housed in a temperature- and humidity-controlled pathogen-free facility with 12-h light/dark cycle and ad libitum access to water and standard laboratory rodent chow.

**Chemical probes**. THZ1, THZ1-R, and YKL-1-116, all of which have been previously described[4,27], were generously provided by Nathanael Gray's lab.

**Mouse models of cardiac hypertrophy and heart failure**. All mice were male C57Bl/6J mice aged 10 to 11 weeks from The Jackson Laboratory. Mice were placed on a temperature-controlled small-animal surgical table to help maintain body temperature (37 °C) during surgery. Mice were anesthetized with 1–3% isoflurane, mechanically ventilated (Harvard Apparatus), and subjected to thoracotomy. To perform transverse aortic artery constriction, aortic arch was exposed and constricted between the left common carotid and the brachiocephalic arteries using a 7-0 silk suture and a 27-gauge needle as previously described[29]. Mice in the sham group were subjected to the same procedures except the placement of constriction. Daily intraperitoneal injections of vehicle solution (0.15 mL of 10% DMSO and 5%

dextrose in water)[39] or 20 mg/kg THZ1 (dissolved in 0.15 mL of the same vehicle solution) were started on postoperative day 1 until postoperative day 60.

**Neonatal rat ventricular myocyte isolation and culture.** Neonatal rat ventricular myocytes (NRVMs) were isolated from the hearts of 2-3 day old Sprague-Dawley (Strain #001) rat pups from Charles River and maintained under standard conditions as described previously[40]. After overnight digestion with 0.25% trypsin at 4 °C, cells were dissociated from the tissue through a series of digestion with 300 U/mL collagenase II (Worthington Biochemical Corporation, Lakewood, NJ). The dissociated cells were then pre-plated for 1.5 h in cell-culture dishes followed by 24 h exposure to BrdU in culture medium to remove contaminating nonmyocytes and inhibit any residual cellular proliferation. Unless otherwise stated, NRVM were plated at a density of $10^5$ cells/mL. Cells were initially plated in growth medium (DMEM supplemented with 5% FBS, 100 U/ml penicillin-streptomycin and 2 mM L-glutamine) for 48 h. Prior to stimulation with agonists or chemical probes, NRVM were maintained in serum-free medium (DMEM supplemented with 0.1% BSA, 1% insulin-transferrin-selenium supplement (Sigma I3146), 100 U/ml penicillin-streptomycin, and 2 mM L-glutamine) for additional 48 h.

For individual CDK knockdowns, *Cdk7, 12, and 13* mRNA were targeted by specific siRNAs (Sigma SASI_Rn02_00333956, SASI_Rn01_00106147, and SASI_Rn02_00272683 respectively) at a final concentration of 50 nM. For combinatorial triple knockdown of *Cdk7/12/13*, equal concentrations of each siRNA probe were adjusted to maintain a final siRNA concentration of 50 nM. Transfection was performed with transfection reagent RNAiMAX (Invitrogen) according to the manufacturer's instruction. Control groups were transfected with MISSION siRNA Universal Negative Control #1 (Sigma) at the same final concentration.

For testing the effects of small-molecule probes in the hypertrophic response, NRVM were incubated with vehicle (DMSO) or inhibitors at indicated concentrations for 3 h followed by stimulation with phenylephrine (PE) (100 μM) for various durations. For mRNA level and cell size measurements, NRVMs were stimulated with PE for 48 h. For evaluating the phosphorylation level of RNA polymerase II, NRVMs were incubated with PE for 30 min before harvesting.

**Human iPSC-CMs culture, qRT-PCR, and BNP enzyme-linked immunosorbent assay (ELISA).** Human-induced pluripotent stem cell-derived cardiomyocytes (iCell cardiomyocytes) were purchased from FUJIFILM Cellular Dynamics, Inc. and maintained according to the manufacturer's instructions. IRB review was not required. Cells were seeded on 96-well culture plates coated with 5 μg/mL of fibronectin and maintained in Williams' E Medium supplemented with 1:25 of Cell Maintenance Cocktail B (Thermo Fisher Scientific, CM4000). For subsequent hypertrophic stimulation, quiesced iPSC-CMs were treated with or without THZ1 at indicated concentrations versus DMSO for 3 h followed by stimulation with either vehicle or 10 nM endothelin-1 (Sigma) for 18 h. The complementary DNA was directly synthesized from RNA in cell lysate using the TaqMan Gene Expression Cells-to-CT kit (Thermo Fisher Sci AM1728) according to the manufacturer's instruction. qRT-PCR for human *NPPB* and *B2M* (normalizer) was performed with the Life Technologies TaqMan assays (NPPB, Hs00173590_m1; B2M, Hs000984230_m1). qRT-PCR for other human genes was performed using TaqMan chemistry including FastStart Universal Probe Master (Roche), labeled probes from the Universal ProbeLibrary (Roche), and gene-specific oligonucleotide primers (list of qRT-PCR primers and TaqMAN probes are provided in Supplementary Data 4). Relative expression was calculated using the $2\Delta\Delta Ct$ method with normalization to *B2M* expression. ELISA for NT-proBNP protein was performed on 6 ml of medium from each well of a 96-well plate as previously published[28] using the following antibodies: anti-proBNP capture antibody (clone 5B6; Abcam ab13111), anti-proBNP HRP conjugate detection antibody (clone 16F3; Abcam ab13124), and purified NT-proBNP (Phoenix Pharmaceuticals, catalog no. 011–42) as a standard.

**Western blotting.** To extract total cellular protein, cells were lysed in RIPA Buffer (Sigma R0278) supplemented with protease inhibitor (Roche cat. no. 4693132001) and phosphatase inhibitor (Phos-STOP; Roche, 04 906 845 001) tablets. Nuclear protein was isolated using the NE-PER Nuclear and Cytoplasmic Extraction Reagents (Thermo Scientific, 78833) according to the manufacturer's instructions. For each sample, 20–50 μg protein extracts were separated by 8–12% Bis-Tris gels (Fisher Sci, NW04122BOX), transferred to nitrocellulose membranes. To ensure uniform protein loading for each antibody signal, all protein samples were prepared as a single batch that was adjusted to the same total protein concentration and pre-boiled in loading buffer. Western blots were run by loading equal volumes of this lysate prep to achieve uniform total protein mass and volume loading in each lane. As we were detecting multiple RNA Pol II phosphoforms which run at overlapping mobility from the same sample, the determination of signal by serial stripping and reprobing of each membrane did not provide appropriate fidelity for reliable signal discrimination. Therefore, in most cases, Westerns were performed for the various Pol II phosphoforms on individually run gels and membranes. The specific membrane from which the loading control is run is designated for each Western panel in the figure legend. Specific targets were detected with the following

antibodies: CDK7 Antibody (C-19) (Santa Cruz sc-529) or CDK7 Antibody (C-4) (Santa Cruz sc-7344), alpha-tubulin (Sigma T9026), RNA Polymerase II (1F4B6) (Active Motif 61667), Ser2P RNA Pol II (Abcam, ab5095) or Ser2P RNA Pol II (3E10) (EMD Millipore, 04–1571), Ser5P RNA Pol II (3E8) (EMD Millipore, 04–1572), Ser7P RNA Pol II (4E12) (EMD Millipore, 04–1570).

**Immunocytochemistry and immunohistochemistry**
*Cultured cells.* NRVMs were seeded on glass coverslips coated with 0.1% gelatin and fixed in 2% paraformaldehyde for 20 min at room temperature followed by permeabilization with PBST plus 0.1% Triton X. NRVMs were incubated in PBST plus 5% horse serum for 1 h and then stained with primary antibody against α-actinin at 1:800 (Sigma-Aldrich, A7811) for 1 h. NRVMs were washed 3 times with PBST followed by staining with secondary antibody at 1:1000 for 1 h. Cells were washed 3 times with PBST and mounted with Vectashield Hardset mounting medium. For human iPSC-derived cardiomyocytes, cells were plated on Ibidi 8-well chamber slides. Cells were fixed in 2% paraformaldehyde for 20 min, permeabilized with PBST plus 0.1% Triton X-100 for 20 min, and blocked with PBST plus 5% horse serum for 1 h at room temperature. Primary antibody against sarcomeric α-actinin was used at a dilution of 1:800 in PBST plus 5% horse serum for 1 h at room temperature. A secondary anti-mouse antibody was used at a dilution of 1:1000 in PBST plus 5% horse serum for 1 h at room temperature. Cell area for NRVM and human iPSC-derived cardiomyocytes was quantified from immunofluorescent images as previously described[29].

*Mouse heart tissue samples.* Mouse hearts were harvested, dried and weighed. Tissue was fixed in 10% neutral-buffered formalin overnight at 4 °C and transferred to 100% ethanol for 24 h. Samples were embedded in paraffin blocks, sectioned, and stained with Picrosirius red (Polysciences) or Wheat Germ Agglutinin 488 (Thermo Fisher Scientific, W11261) according to the manufacturers' instructions. Images were captured by fluorescence microscopy, bright field microscopy or high-content imaging platform (IN Cell Analyzer, GE Healthcare Systems) as previously described[29]. For cardiomyocyte cross-sectional area from WGA-rhodamine stained LV tissue samples, we quantified area of 50 randomly selected cardiomyocytes per heart from 3 randomly chosen fields at 200x magnification (from tissue sectioned from the mid-papillary region in the LV short-axis), using Image J. This was repeated in 3 different hearts for each experimental group, leading to a total of 150 individual cardiomyocyte cross-sectional area measurements that were pooled for each experimental condition. Quantification of LV fibrosis from picrosirius red-stained heart tissue sections was performed in 5 independent hearts per experimental group as previously described[29].

**Echocardiography and blood pressure measurements.** For transthoracic echocardiography, mice were anesthetized with 1-2% isoflurane and imaged using Vevo 770 Imaging System (Visual Sonics, Inc.) with RMV-707B probe. Measurements were obtained from M-mode sampling and integrated electrocardiogram-gated kilohertz visualization (EKV) images taken in the LV short axis at the mid-papillary level to generate high spatiotemporal resolution two-dimensional images. LV areas and ejection fraction were obtained from high-resolution two-dimensional measurements at end-diastole and end-systole[29].

Non-invasive blood pressure was measured using the BP2000 Blood Pressure Analysis System (Visitech Systems, Inc.) according to the manufacturer's instruction. Mice received chronic injections of vehicle or THZ1 (20 mg/kg/day) injections for 1 week. During this period, mice were trained on the Visitech system to enable accurate, standardized BP measurements. Blood pressure was recorded at 24 h after the most recent injection on 3 different days for each mouse. Data were then pooled and analyzed.

**RNA isolation and qPCR.** Total RNA was isolated from NRVM using the High Pure RNA isolation kit (Roche 11828665001) with on-column DNAase treatment according to the manufacturer's instruction. For RNA isolation from mouse tissue, a 10–20 mg piece of mouse heart ventricles was mechanically homogenized in PureZOL RNA isolation reagent (Bio-Rad 7326890) on a TissueLyser II (QIAGEN 85300) using stainless steel beads (QIAGEN 69989). Total RNA was purified using the Aurum total RNA purification kit (Bio-Rad 732–6830). The purified total RNA was then reverse transcribed to complementary DNA with iScript RT supermix (Bio-Rad 170–8841) according to the manufacturer's instruction. A list of qRT-PCR primers and TaqMAN probes are provided in Supplementary Table 3. Relative expression was calculated using the $2\Delta\Delta Ct$ method with normalization to constitutive genes. Mouse data were normalized to *Ppib*.

**Library preparation and next-generation sequencing for RNA-Seq.** RNA isolated from tissue and cells were quantified using Qubit fluorometric quantitation and assessed for quality using Agilent Bioanalyzer Nano RNA Chip. Samples with RNA integrity number larger than 8 were considered high-quality and suitable for RNA-seq. Library preparation was performed using the Illumina TruSeq Stranded mRNA kit or KAPA mRNA HyperPrep Kit (Roche), and the library was sequenced on an Illumina HiSeq 2500 at the Genomics Core Facility of Whitehead Institutes (single-end, 40 base pairs, >50 M reads per sample).

**RNA-seq data processing and quantification**. Each RNA-Seq sample was aligned independently. Specifically, the raw RNA-Seq reads from each sample were aligned with TopHat v2.1.1 with options "-p 8 -r 300–library-type fr-firststrand–no-coverage-search". Each sample was aligned either against the mm9 or rn5 reference genome as appropriate for the sample.

After alignment, the number of mapped reads per gene per sample using *htseq-counts* was quantified. Each application of *htseq-counts* was run with the following options:"-s reverse –f bam". Each sample was quantified against the GENCODE mm9 transcript reference or the UCSC rn5 transcript reference, depending on the source of the sample.

The differentially expressed genes were identified by DeSeq2. The quantifications from htseq-count were imported into DeSeq2, and all samples from the same organism were analyzed together in one analytical run. Replicate samples were grouped by their type (e.g., baseline, THZ1-treated) in order to better estimate the sample-level variance for comparison across samples. The standard DeSeq2 workflow followed without modifications, and Benjamin-Hochberg correction was used to control false discovery of differentially expressed genes.

Principal component plots were generated from a matrix of the normalized counts from DeSeq2. Normalized counts were re-centered and scaled per sample prior to the analysis in order to standardize each sample. Heatmaps were generated from the normalized DeSeq2 counts. After row-scaling, individual rows were clustered using hierarchical clustering with a Euclidean distance metric and Ward's clustering. Bar graphs were generated using calculated fold changes of expression of representative genes from DeSeq2 and the adjusted *p*-values reported by the software. The numbers of genes in the Venn diagram were generated using adjusted *p*-value < 0.05 (5% FDR) and Log2 fold change ≥ 1 for comparison between the TAC-Veh group vs Sham-Veh group, and adjusted *p*-value < 0.05 (5% FDR) and Log2 fold change ≤ −0.89 for comparison between TAC-Veh group vs TAC-THZ1 group. For the TAC samples, the fold change cutoff of ≤ −0.89 was chosen to capture the differential expression of *Nppb* between TAC-veh vs. TAC-THZ1 in this *n* = 4 sample set, so as to maintain a threshold that appropriately captured a known positive control gene.

DAVID (v.6.8)[41] was used to identify enriched functional annotations in differentially expressed gene ID lists relative to the set of "expressed" genes, defined as having sufficient normalized counts for DESeq2 to calculate a differential expression *p* value, or versus all annotated genes.

For gene set enrichment analysis (GSEA), command line GSEA2 (v.2.2.2, ref. [5].) was used with MSigDB v.6.1 gene sets[40]. Values were pre-ranked by log2 fold change, retaining the absolute maximum change per gene where multiple transcripts were assessed. Significance was expressed as family-wise error rate (FWER). FWER *p* < 0.250 represents statistically significant enrichment.

**Chromatin immunoprecipitation, library preparation and next-generation sequencing**. For ChIP experiments, NRVM were plated in 15 cm dishes at 4 ×10^6 cells/dish. Chromatin derived from a total of 15 million cells were used for each immunoprecipitation. After indicated treatments, cells were crosslinked with 1% formaldehyde for 10 min at room temperature followed by quenching with 0.125 M glycine for 5 min. Cells were then lysed and chromatin was extracted. Isolated chromatin was subjected to shearing with Bioruptor (Diagenode) for 16 cycles (30 sec on and 30 sec off/cycle) at high intensity. A small volume of sheared chromatin (50 uL) from each was stored at −80 °C as input. The rest of sheared chromatin was then incubated with 5 mg RNAPII antibody (clone 8WG16, Bio-Legend 920102 lot B217159) with 50uL Dynabeads (Invitrogen) overnight at 4 °C. Chromatin was washed, eluted and reverse crosslinked overnight at 65 °C. Genomic DNA was purified from both immunoprecipitated samples and input samples. The library was prepared with TrueSeq ChIP kit (Illumina), and the library was sequenced on an Illumina HiSeq 2500 at the Genomics Core Facility of Whitehead Institutes (single-end, 40 base pairs, >50 M reads per sample).

**ChIP-Seq data processing**. Each ChIP-Seq sample was processed and aligned using bowtie 1. Each sample was aligned against the rn5 reference genome using bowtie with options "-k 1 -m1". All gene track and ChIP-seq quantifications were normalized by the number of (uniquely) mapped reads to estimate the "reads per million" (rpm) quantity. Gene track displays were made with Integrative Genome Viewer (IGV).

All density plots for Pol II were made using all rn5 gene promoters based on the rn5 UCSC transcription annotation. Each density plot estimated the average Pol II ChIP-seq signal per promoter within a ±5 kb window.

Traveling ratios for genes that are differentially expressed were estimated as previously decribed[31]. The PE-responsive differentially expressed genes used in traveling ratio analysis were identified in RNA-Seq data analysis by comparing the PE group to the baseline group with an adjusted *p*-value cut-off of 0.1 (10% FDR).

**Dissecting the gene expression changes elicited by THZ1 in diverse cardiac cellular compartments by comparing our bulk RNA-seq data with published scRNA-Seq datasets**. To dissect compartment-specific gene expression changes from our bulk RNA-seq data from LV tissue of THZ1-treated mice in the TAC model, scRNA-Seq data performed independently by our group in the mouse TAC model (TAC and sham groups were used) using 10X Genomics were curated from

Alexanian et al.[34]. To recover more comprehensive cardiac cell types, including adult cardiomyocytes, and to increase the sensitivity of gene detection, we also downloaded and curated mouse heart scRNA-Seq by Smart-seq2 from the Tabula Muris cohort[35]. These two scRNA-seq datasets were filtered, normalized, and integrated by Seurat (v4.0.0) R package[42] with canonical correlation analysis. After performing dimensional reduction by Principal Component Analysis (PCA), the top 30 PCAs were used in graph-based clustering based on Louvain with resolution at 0.5, and cluster-specific marker genes were identified using the FindAllMarkers function by Wilcoxon Rank Sum test in Seurat. Cell types were determined by cross-referencing with source data papers and human heart atlas references[34,35,43,44]. Uniform Manifold Approximation and Project (UMAP) were used to visualize the high-dimensional cell cluster distribution. Cell-type enriched signature genes were identified using the FindAllMarkers function by Wilcoxon Rank Sum test in Seurat. Cell type enriched genes that were differentially expressed in TAC vs Sham by bulk expression analysis were selected for downstream analysis. TAC altered cell-type enriched signature genes were visualized in a graphic heatmap using the ComplexHeatmap (v2.6.2) R package[45].

**Statistical analysis**. Measurements were taken from distinct samples. Data are reported as mean ± SEM unless otherwise indicated in the figure legend. Statistical analysis of LV fractional shortening as a function of time was performed by ANOVA with Holm-Sidak correction for multiple comparisons using GraphPad Prism. All other statistical analyses were performed using ANOVA with Tukey's honest significant difference test correction for multiple comparisons using GraphPad Prism. For all analyses, *p* < 0.05 was considered significant. The statistical methods used in the analyses of RNA-seq and ChIP-seq data are detailed separately above.

## Data availability

The data supporting the findings from this study are available within the manuscript and its supplementary information. The RNA-seq datasets generated in this study have been deposited in the Gene Expression Omnibus (GEO) database under accession codes GSE151253[46] (NRVM RNAseq) and GSE151254[47] (Mouse heart RNAseq). The ChIP-seq datasets generated in this study have been deposited in the GEO database under accession code GSE151252[48] Source data are provided with this paper.

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

## Acknowledgements

B.G.B. acknowledges support by the Gladstone Institutes and the Younger Family Fund. This work was supported by NIH grants HL127240 (to S.M.H. until September 2018), GM123511 (R.A.Y.), and 5 U01 CA213333-03 (N.S.G.). A.H. was supported by a pre-doctoral fellowship from the Tobacco-Related Disease Research Program (28DT-008), the UCSF Discovery Fellows Program, an Achievement Rewards for College Scientists Scholarship, and NIH grant T32GM008568. A.P. is supported by a Scholar Award from the Sarnoff Cardiovascular Foundation, NIH K08 HL157700, Tobacco-Related Disease Research Program (578649), A.P. Giannini Foundation (P0527061), and Michael Antonov Charitable Foundation Inc. M.A. is supported by the Swiss National Science Foundation (P400PM_186704).

## Author contributions

A.H., Q.D., D.S.D., B.G.B., and S.M.H. conceived the study, interpreted the data, and wrote the manuscript. Y.H. performed heart surgeries and echocardiography. Z.B.F. performed ChIP-seq. A.H. and Q.D. performed primary cell isolations for knockdown and chemical inhibitor experiments and performed molecular and biochemical analyses on in vivo samples. S.M. performed human iPSC-CM studies. T.Z., Y.L., and N.S.G. provided chemical compounds and provided expertise and guidance on studies. D.S.D. performed RNA-seq and ChIP-seq computational analyses. X.L. performed integrated bulk and single-cell RNA-seq analysis. Z.J., M.A., A.P., J.D.B., C.Y.L., and R.A.Y. provided expertise, advisement, and critical revisions to the manuscript.

## Competing interests

S.M.H. is an officer, executive and shareholder of Amgen, a scientific founder and shareholder of Tenaya Therapeutics, and serves on the Scientific Advisory Board of the DZHK (German Centre for Cardiovascular Research) of the German Ministry of Health as an uncompensated volunteer. X.L. is an employee and shareholder of Amgen. B.G.B. is a scientific founder and shareholder of Tenaya Therapeutics. R.A.Y. is a founder and shareholder of Syros Pharmaceuticals, Camp4 Therapeutics, Omega Therapeutics, and Dewpoint Therapeutics. N.S.G. is a Scientific Founder, member of the SAB and equity holder in C4 Therapeutics, Syros, Soltego (board member), B2S/Voronoi, Allorion, Lighthorse, Cobroventures, GSK, Larkspur (board member) and Matchpoint. The Gray lab receives research funding from Springworks and Interline. T.Z. is a scientific funder, equity holder and consultant of Matchpoint. The remaining authors declare no competing interests.
