## [Peer Review File · Nature Communications]

Targeting Transcription in Heart Failure via CDK7/12/13 inhibitionREVIEWER COMMENTS

Reviewer #1 (Remarks to the Author):

The authors identified the critical regulator of transcription elongation CDK7 as a mediator of cardiomyocyte hypertrophy. They identified a chemical, THZ1, which by inhibiting CDK7 prevents cardiac hypertrophy in vitro and in vivo. The manuscript is interesting. I have the following questions.

- Almost two decades ago, Michale Schneider and his group published that CDK9, not CDK7, is important for cardiac hypertrophy in vivo and in vitro. The data Schneider's group published seem still now quite compelling. This is a major issue the authors need to address as it affects the novelty and the meaning of their observation significantly.

- The authors rely on the THZ1 compound for inhibiting CDK7. As a proof of principle a cardiac-specific KO should be used to determine whether in vivo CDK7 is relevant for cardiac hypertrophy and failure. As a matter of facts, extra-myocardial effects of CDK7 cannot be excluded.

Alternatively, CDK7 activity should be measured, with or without THZ1, in different myocardial cell types during TAC.

Minor:

-In fig.1 C, the decrease of the phosphorylation levels of Pol II at Ser 2, 7 and 5 after CDK7 depletion are not evident. This data set should be improved.

-In fig. 1G, author show that THZ1, causes a decrease of phosphorylation at Ser 2, 5 e 7. If Ser 5 and 7 are CDK7 target, Ser 2 is a target of CDK9. The authors should explain this result.

- Authors should provide evidence that CDK7 activity changes during hypertrophy in vivo, since Sano et al demonstrate that CDK7 does not change in mice after TAC and in rodent CMs stimulated with ET-1.

- Is the effect of THZ1 during TAC transient or continuous? What is the effect on transcription at different time-points?

. Authors could carry out ChIP-seq experiments for Ser7p and calculate the pol II pausing index (the ratio of pol II density at the promoter/gene body), which should be modified with CDK7 inhibition.

Reviewer #2 (Remarks to the Author):

This submission explores whether inhibition of Cyclin-Dependent Kinase 7 (CDK7), the catalytic part of the trimeric CAK complex, can attenuate the pathological effects of stress-induced transcription and hypertrophy during the development of HF. It is an essential component of the transcription factor TFIIH, that is involved in transcription initiation and DNA repair. The manuscript is highly novel in that it demonstrates a role for CDK7 in initiation of maladaptive transcriptional initiation, and that pharmacological targeting upstream of genes such as Nppa and Nppb may be a potent method of attenuating progression to HF. A major strength of this submission is that the oncological data supporting THZ1 as a potent regulator of pathological gene transcription via CDK7 inhibition are already well-established, with safety and efficacy studies well-documented in cancer models. This submission presents a repurposing of a cancer drug for HF, which would expedite testing in this patient population. Overall, the manuscript is of high to exceptional quality, with a broad readership due and high clinical relevance. All experiments performed were appropriate with proper statistical analysis and controls. Data are compelling and well-organized to support the premise. Some questions from the reviewer as well as suggestions to improve readability and completeness are detailed below.

Reviewer comments:

The authors state that "Like THZ1, YKL-1-116 inhibited cellular hypertrophy and regulation of Nppa/Nppb induction at concentrations similar to those that inhibit cancer cell states." But in comparing the IC50s of these two compounds (Fig.1F to Supp.Fig. 2B), THZ1 was tested on NRVM at doses of 1-10 nM but YKL-1-116 at 0.1-5.0 μ M (100-5,000 nM) – a huge difference in functional IC50s. Which dose range is more comparable to oncologically therapeutic doses? Reviewer assumes both cannot be correct. The authors cannot make a claim that these two compounds are equivalent/comparable and should instead temper their claim to state that both are effective. However, the use of two structurally distinct pharmacological inhibitors to illustrate the importance of CDK7 and support the RNAi data in abrogating PE-induced hypertrophy was an appropriate and well-utilized approach. Suppl.Fig.1B shows low-mid nM doses for various cell lines

In Fig.2B, authors should at least briefly detail any common pathways or functions for the genes in Groups 1 and 3 which were altered at baseline with THZ1 treatment. Of note, hsp90, a critical cardiac chaperone protein, is listed. These gene clusters may represent important basal effects on cardiomyocytes which are not assessable via hypertrophic measurement (e.g., contractility, Ca²⁺ flux, UPR).

TAC (60d +/- CDK7i) as a model of HFrEF is appropriate and the rescue effects are marked. In Supp.Fig.3G, why was Pol II Ser7P only measured? Were there no changes to Pol II Ser2P/Ser5P in the TAC mice without pharmacological inhibition?

Authors assessed overall body weight of TAC and sham mice treated with THZ1, but were cachexia or cell death in skeletal or SMC myocytes measured? The question arises whether long-term clinical administration of THZ1 would have detrimental effects on other cell types (particularly those with high turnover) due to its transcriptional inhibitory effects. The authors state that at the doses used, "THZ1... has been validated to provide target coverage and efficacy in murine tumor xenograft studies and is well tolerated in mice," but do not mention whether patients on THZ1 therapy for tumor subtypes experience any side effects. Of note, CDK7 is important for neuronal plasticity. Reviewer suggests that some discussion about the feasibility of administration to HFrEF patients be included, particularly any potential drawbacks.

Reviewer #3 (Remarks to the Author):

In this study, the authors investigate a role for the transcriptional kinase Cdk7 in pathogenesis of heart failure with reduced ejection fraction (HFrEF), and demonstrate potential therapeutic efficacy of inhibitors targeting this kinase in preclinical models. They show in a rat cardiomyocyte culture model that depletion of Cdk7 by RNAi can limit hypertrophy and attenuate pathological gene expression changes induced by the α 1-adrenergic agonist phenylephrine (PE), and replicate these findings with the covalent inhibitor THZ1, which has moderate selectivity for Cdk7. They validate these effects in vivo in a mouse model of heart failure, and in human iPSC-derived cardiomyocytes. There are some mechanistic questions I would like to see addressed, and some methodological issues I would hope to see clarified, before recommending publication, but overall this is a well-executed study, and the results have potential therapeutic applications to a major human health problem. My specific concerns and questions:

1. In Fig. 1A, the si-Cdk7 cells appear larger than si-cntrl in the absence of PE. Is this difference significant (statistically) and reproducible? If so, can the authors comment on what it might signify?
2. The effects of Cdk7 knockdown on Pol II CTD phosphorylation in PE-treated cardiomyocytes are quite modest (Fig. 1C) and should be quantified. It should also be pointed out that CTD phosphorylation changes due to Cdk7 knockdown (or inhibition) are likely a mix of direct and indirect effects (i.e., reduced transcription leading to decreased overall Pol II phosphorylation); Cdk7 is not thought to phosphorylate Ser2, and as the authors point out, multiple kinases can modify Ser5 and Ser7 in vivo.
3. On a related topic, the most dramatic effect of THZ1 in this system is on Ser2 phosphorylation,

which might reflect the indirect mechanism invoked in point 2 (and by the authors later in the text) but could also be due to the significant potency of THZ1 towards Cdk12 and Cdk13, two transcriptional kinases that have been directly implicated in Ser2 phosphorylation (and are likely responsible for some of THZ1's anti-cancer effects—see for example, ref. 36). That is probably the case for the potentially therapeutic effects described here: Note that the doses of the more Cdk7-selective inhibitor YKL-1-116 required to block cell hypertrophy or prevent induction of Nppa and Nppb are higher than those of THZ1 (cf. Supp. Fig 2B,C and Fig. 1E,F) even though YKL-1-116 is at least as potent an inhibitor of Cdk7 (see ref. 29). To define Cdk7's role in HFrEF with any precision (or the potential efficacy of Cdk7-selective inhibition in ameliorating it), more extensive experiments with more selective compounds such as YKL-1-116 would be needed. In the meantime I suggest the authors tone down or qualify some of their stronger statements about Cdk7 being the central transcriptional CDK in HFrEF pathogenesis, or the primary therapeutic target of THZ1 (e.g. in the title).

4. A minor point, but I would recommend the authors provide a brief description in the main text of how they calculated "traveling ratio" (Fig. 2E). Unfortunately, this term has been used in the past to refer to either the ratio of Pol II over the TSS versus the gene body (which has also been called the "pause index"), or its reciprocal ratio (i.e., gene body over TSS). Although the former definition ($TR=PI$) seems to have won out, and appears to be the one used here, it would help to say so explicitly to avoid ambiguity.

5. This result is somewhat surprising and potentially interesting, but requires some context and additional explanation. Previous analyses of Pol II occupancy after Cdk7 inhibition (or THZ1 treatment) indicated decreased density across gene bodies with, if anything, greater depletion over the TSS (see for example ref. 3). That pattern predicts an unchanged or decreased PI/TR (see point 4) and is consistent with the role of Cdk7 activity in recruiting the pausing factors DSIF and NELF to establish the promoter-proximal pause (Glover-Cutter et al., *Mol. Cell Biol.* 29:5455, 2009; Larochelle et al., *Nat. Struct. Mol. Biol.* 19:1108, 2012). Can the authors comment on whether this behavior is specific to the genes induced by PE (i.e., does the TR go up or down in the cluster 1 genes repressed by THZ1 treatment in the absence of PE)? Another potential source of the difference with previous studies is the time of treatment with THZ1, but I could not find that information for the RNA-seq or ChIP-seq experiments. Was it the 48-hr time of treatment used in Fig. 1E? (If so, that could explain the discrepancy since the previous studies used much shorter durations of inhibition.)

6. In the PCA (Fig. 4B), is there any explanation for the large shift along PC2 in 2 of the 4 mice that received TAC + THZ1?

Reviewer #1 (Remarks to the Author):

The authors identified the critical regulator of transcription elongation CDK7 as a mediator of cardiomyocyte hypertrophy. They identified a chemical, THZ1, which by inhibiting CDK7 prevents cardiac hypertrophy in vitro and in vivo. The manuscript is interesting. I have the following questions.

- Almost two decades ago, Michael Schneider and his group published that CDK9, not CDK7, is important for cardiac hypertrophy in vivo and in vitro. The data Schneider's group published seem still now quite compelling. This is a major issue the authors need to address as it affects the novelty and the meaning of their observation significantly.

We thank the reviewer for highlighting this important point. We agree that the data from Michael Schneider's seminal study, which identify CDK9 (the core kinase in the PTEFb complex) as a positive regulator of cardiomyocyte hypertrophy (Sano *et al.* Nature Medicine 2002)¹, are indeed important. However, we note several key differences between the experimental approaches and conclusions of Michael Schneider's paper and those of our current manuscript, which we discuss below. These differences highlight that the findings in our study are both novel and compatible with those reported by Sano *et al.*¹.

In Sano *et al.*, the authors used the pleiotropic chemical inhibitor Benzimidazole (DRB) at micromolar concentrations (50uM) to inhibit CDK9 kinase activity, a concentration that has promiscuous effects. For example, the published IC₅₀ for DRB against CDK9 and Casein Kinase II are both in the micromolar range (3uM and 4-10uM respectively)²⁻⁵ and Casein Kinase II was recently shown as an important mediator of cardiac hypertrophy⁶. The observed effects of DRB at this concentration were likely due to pleiotropic inhibitory effects on several kinases. The authors did not utilize specific CDK9 knockdown or knockout approaches to corroborate their chemical inhibitor data, but rather demonstrated anti-hypertrophic effects with adenoviral overexpression of a dominant negative CDK9 (dnCDK9) mutant. We interpret these data to support a role for CDK9 and the PTEFb complex in regulating cardiac hypertrophy, but they do not exclude a role for other important transcription kinases such as CDK7, 12 and 13 in this process.

Importantly, the study by Sano *et al.*¹ did suggest a partial role for CDK7 in cardiomyocyte hypertrophy. They reported that adenoviral overexpression of a dominant negative CDK7 mutant in cultured cardiac myocytes partially suppressed [³H]uridine and [³H]phenylalanine incorporation during endothelin-1 stimulated hypertrophy (though to a lesser extent than that of dnCDK9). Furthermore, Sano *et al.*¹ did not test the role of CDK7 or CDK9 inhibition in vivo but rather demonstrated that enforced overexpression of Cyclin T1, an upstream activator of CDK9 activity, can induce cardiac hypertrophy in mice when chronically expressed from birth at high levels.

We emphasize that the fundamental difference between Sano *et al.*¹ and our current study is that the former focused on the role of the PTEFb complex (in which the core kinase is CDK9). By contrast, our study centers on the TFIIH transcription regulatory complex, in which CDK7 is the core kinase with some overlapping contribution from two closely related kinases, CDK12 and CDK13. A mounting body of evidence now supports that CDK12 and CDK13 can indeed compensate for CDK7 when the latter is inhibited^{7,8}. This functional redundancy is supported by our observation that siRNA-mediated knockdown of CDK7 does not universally and consistently recapitulate the potent effects of THZ1 on all aspects of cardiomyocyte hypertrophy, suggesting that CDK12 and CDK13 may be compensating in this setting (Figure 1 and Supplementary

Figure 1). It should be noted that the chemical probes used in this manuscript not directly inhibit CDK9^{8,9}, supporting a specific role for CDK7/12/13 in cardiac hypertrophy.

Therefore, in the revised manuscript we no longer focus exclusively on CDK7 inhibition and have performed new experiments that completely reframe the narrative around the combined effects of CDK7/12/13 inhibition, which more accurately reflects the pharmacology of the small molecule THZ1. First, we have added an extensive series of experiments in which we perform individual knockdown of *Cdk7*, *Cdk12*, and *Cdk13* in NRVM to illustrate that each manipulation alone does not consistently recapitulate the potent effects of THZ1 on cardiomyocyte hypertrophy (see panel below; these data now included in Supplementary Figure 1 of the revised manuscript):

We also perform simultaneous triple knockdown of *Cdk7/12/13*, a manipulation which more closely recapitulates the pharmacology of THZ1, to demonstrate a more robust and consistent effect on cardiomyocyte hypertrophy and Pol II phosphorylation (see panel below; these data are now included in Figure 1 of the revised manuscript).

Based on these considerations, we believe our revised manuscript supports the novel concept that combined inhibition of CDK7/12/13, either using genetic knockdown or the potent small molecule inhibitor THZ1, potentially inhibits cardiomyocyte hypertrophy. Our unbiased transcriptomic and genome wide Pol II occupancy data in THZ1-treated NRVM (Figure 2) also support our mechanistic conclusions that THZ1 blocks stress-dependent transcription during hypertrophic stimulation of NRVM. Overall, we believe that these findings pertaining to CDK7/12/13 are novel and compatible with the observations pertaining to CDK9/PTEFb by Sano *et al*¹.

We have now emphasized the ability of THZ1 to inhibit CDK7/12/13 (and not just CDK7) throughout the manuscript. We have also added a Discussion paragraph to discuss our findings in the context of the publication by Sano *et al* .:

“Previous studies demonstrated that the CDK9/PTEF-b complex played an important role in cardiac hypertrophy⁵. Using pleiotropic chemical inhibitors such as flavopirodol or adenoviral overexpression of a dominant-negative CDK9 construct, CDK9 inhibition was shown to attenuate cardiomyocyte hypertrophy and Pol II hyperphosphorylation in cultured neonatal rat cardiomyocytes in vitro. In addition, cardiomyocyte-specific transgenic overexpression of Cyclin-T1, a key component of the CDK9-activating complex, was sufficient to drive pathological cardiac hypertrophy in mice. These seminal findings suggested a partial role for the CDK9/PTEF-b transcriptional complex in cardiac hypertrophy, although the therapeutic potential of Pol II kinase inhibition could not be assessed due to lack of potent and specific chemical probes that were suitable for in

vivo administration. Interestingly, adenoviral overexpression of a dominant negative CDK7 construct was also shown to modestly attenuate agonist-induced protein synthesis in cultured cardiomyocytes⁵, suggesting parallel approaches to interdict transcription during cardiomyocyte stress responses. Our study demonstrates that simultaneous inhibition of CDK7, 12 and 13 dependent transcription is required for potent antihypertrophic responses and establishes proof-of-concept that CDK7/12/13 inhibition using THZ1 can ameliorate pressure overload induced heart failure in adult mice.”

- The authors rely on the THZ1 compound for inhibiting CDK7. As a proof of principle a cardiac-specific KO should be used to determine whether in vivo CDK7 is relevant for cardiac hypertrophy and failure. As a matter of facts, extra-myocardial effects of CDK7 cannot be excluded. Alternatively, CDK7 activity should be measured, with or without THZ1, in different myocardial cell types during TAC.

The reviewer brings up a very important point about systemic THZ1 administration in mice and the effects it can have on multiple tissues. We also contemplated the use of a mouse harboring a conditionally-targeted *Cdk7* allele. However, new experiments and analyses provided in the revised manuscript support that cell-specific deletion of *Cdk7* alone is unlikely to recapitulate the effects of THZ1. As detailed in the response above, we now show that CDK7, 12 and 13 play redundant roles in cardiomyocyte hypertrophy and that triple knockdown is required to attenuate the hypertrophic response. Therefore, conditionally targeting all 3 of these loci will likely be required to approximate the effect of THZ1 in vivo.

In addition to the issue of targeting multiple CDKs, we also suspected that THZ1 might exert its therapeutic effects in vivo by altering gene expression in multiple cellular compartments in the heart, in addition to cardiomyocytes. While our qPCR data and RNA-seq data from bulk LV tissue demonstrate that THZ1 attenuates stress-mediated upregulation of several cardiomyocyte-specific genes (e.g. *Nppb*, *Acta1*), we also observed that THZ1 affects genes known to be enriched in other cellular compartments in the adult heart, including fibroblasts and immune cells. To more incisively interrogate cell-compartment specific effects of THZ1 in the heart, we have now taken our bulk tissue RNA-seq data and performed extensive “fingerprinting analysis” using curated single cell RNA sequencing datasets from the TAC model that have been recently published by our group¹⁰ and others¹¹. Using these single cell data, we can more precisely assess the cellular compartments that are affected by THZ1 in the TAC model (these data are now included in Figure 4 and Supplemental Figure 6 of the revised manuscript). We observed that in addition to effects on cardiomyocytes, THZ1 caused simultaneous cell state shifts in several other compartments, including fibroblasts, myeloid cells and epicardial cells. A panel detailing these fingerprinting analyses has been added to Figure 4 and is shown below. Representative heatmaps of differentially expressed genes that map to each cellular compartment are provided in Supplementary Figure 6 of the revised manuscript.

These shifts in cell state across multiple cellular compartments suggest that THZ1 can exert on-target transcriptional effects on multiple cell types in vivo that together contribute to its cardioprotective effects in the TAC model. Therefore, attempting to approximate the effect of THZ1 using conditional deletion approaches will not only have to target multiple CDKs, but will also have to target multiple cellular compartments in vivo. This is another reason why conditional deletion of CDK7 in a single cell type is unlikely to recapitulate the therapeutic effect of THZ1 and reflects some of the advantages of leveraging a chemical biological approach. We certainly recognize the limitations of systemic administration of THZ1 to mice and have tempered any conclusions about cell-specific effects on cardiomyocytes in vivo. In addition to the new cell-fingerprinting data that is added to the revised manuscript, we added additional text to our Discussion section to elaborate on the effects of THZ1 on non-myocyte cell compartments:

“While THZ1 can potently suppress agonist induced hypertrophy in cultured cardiomyocytes, our transcriptomic profiling of adult mouse hearts demonstrates that the protective effects of THZ1 in vivo are associated with dynamic gene expression changes in both cardiomyocyte and non-cardiomyocyte compartments. Often, dissecting the

causality of cell specific contributions during heart failure pathogenesis can be interrogated using conditional gene deletion approaches. However, modeling the pharmacology of THZ1 in vivo using conditional gene-deletion approaches will be particularly challenging because it would likely require simultaneous postnatal deletion of multiple loci (Cdk7, Cdk12 and Cdk13). Furthermore, it is expected that gene deletion in any one cellular compartment would only give a partial protection from heart failure pathogenesis and that simultaneous CDK7/12/13 inhibition in multiple cell types, as occurs with THZ1, is required for full therapeutic effects. Finally, in contrast to the intermittent and dose-titratable effects of a chemical inhibitor like THZ1, permanent gene deletion of these 3 kinases may not be as well tolerated. While there are certainly caveats to interpreting the cell compartment and gene-specific effects of THZ1 in vivo, this study highlights some of the advantages of probing disease pathobiology using potent chemical tools.”

Minor:

-In fig.1 C, the decrease of the phosphorylation levels of Pol II at Ser 2, 7 and 5 after CDK7 depletion are not evident. This data set should be improved.

We agree with the reviewer that this data can be improved. We have repeated the western blots for Pol II phosphoforms after CDK7 knockdown NRVM in an independent experiment (shown above in response to the previous comment; included in Supplemental Figure 1E of the revised manuscript). We reproducibly find that CDK7 knockdown results in attenuation of Pol II C-terminal domain phosphorylation, although the effects are not as potent or consistent as nanomolar concentrations of the THZ1 compound. As discussed above, the effect of selective CDK7 knockdown is not as potent as THZ1 likely because CDK12/13 can compensate in this setting. To better approximate the potent pharmacological effect of THZ1, we also performed triple knockdown of CDK7/12/13 and demonstrated a more robust inhibition of Pol II CTD phosphorylation (as discussed above; now included in Figure 1 of the revised manuscript). These data support that the full potency of THZ1's anti-hypertrophic effects are due to combined inhibition of CDK7/12/13 activity.

-In fig. 1G, author show that THZ1, causes a decrease of phosphorylation at Ser 2, 5 e 7. If Ser 5 and 7 are CDK7 target, Ser 2 is a target of CDK9. The authors should explain this result.

The reviewer is indeed correct that Ser2 phosphorylation of the Pol II CTD is not a primary target of CDK7. The observed reduction in Ser2P abundance, which reflects elongating Pol II, may reflect secondary effects of inhibiting upstream steps in the Pol II transcription cycle. Inhibition of these upstream steps in transcription can ultimately reduce the abundance of elongating Pol II, as has been described in prior work¹². In addition, we note that CDK12, which is also a key target of THZ1, has been previously demonstrated to phosphorylate Pol II Ser2P both in vitro and in vivo^{13,14}. We have included new text in the manuscript to highlight the interdependence of Pol II CTD phosphorylation sites and the additional effects of THZ1 on CDK12 and 13.

“THZ1 also inhibits two related Pol II CTD kinases, CDK12 and CDK13, by covalently binding accessible cysteine residues that are homologous to C312 of CDK7”

“THZ1 attenuated the PE-dependent increase in Ser2P and Ser7P (Figure 1C). We did not observe a significant effect on bulk abundance of Ser5P in cardiomyocytes, which may reflect redundant regulation of this phosphoform by other Pol II CTD kinases¹⁵⁻¹⁸.”

“To validate for our findings using THZ1, we also assessed the anti-hypertrophic effect of the small molecule probe YKL-1-116¹⁹ (Supplemental Figure 3A), which has lower potency against CDK12/13. YKL-1-116 inhibited cellular hypertrophy and Nppa/Nppb induction, but required concentrations 50-100 fold higher than THZ1, suggesting that combined inhibition of CDK7/12/13 is required for maximal suppression of hypertrophic stress response in cardiomyocytes.”

- Authors should provide evidence that CDK7 activity changes during hypertrophy in vivo, since Sano et al demonstrate that CDK7 does not change in mice after TAC and in rodent CMs stimulated with ET-1.

Concordant with the observations of Sano *et al.*¹, CDK9 or CDK7 abundance at the mRNA or protein level does not change during cardiac stress. We have also found that bulk CDK7 abundance is not increased in PE-stimulated NRVM (provided below; shown as Supplementary Figure 1C and 1K of the revised manuscript):

Therefore, immunoprecipitation of CDK7 in cell/tissue lysates followed by assessment of in vitro kinase activity in a cell-free system is unlikely to demonstrate differences in kinase activity, because such assays are heavily dependent on the bulk abundance of CDK7 and do not assess the endogenous activity of the transcriptional complex in the context of an intact cell, which can be influenced by chromatin localization, compensation by CDK12/13, and presence of other participants this multi-protein complex. In contrast, the ability of CDK7/12/13 inhibition (using siRNA or potent and specific chemical probes like THZ1) to block hyperphosphorylation of the Pol II CTD directly reflects the endogenous activity of CDK7/12/13-containing transcription complexes in the context of the intact cell. Our NRVM data demonstrate that Pol II CTD phosphorylation is increased with hypertrophic stress and can be inhibited by either CDK7/12/13 knockdown or chemical inhibition with THZ1 (as discussed above; data now in Figure 1 of the revised manuscript). In addition, we demonstrate that Pol II CTD phosphorylation of Ser2P and Ser7P is increased in mouse heart lysates after TAC and is inhibited with administration of THZ1 in vivo (shown above; included as Supplementary Figure 4G of the revised manuscript).

Collectively, these siRNA and pharmacological data support that the net activity of CDK7/12/13 is *specifically* increased during hypertrophic stress and is required for pathological remodeling.

- Is the effect of THZ1 during TAC transient or continuous? What is the effect on transcription at different time-points?

The daily THZ1 dosing strategy utilized in prior studies⁹ as well as our current study suggest that the effect of THZ1 is transient and requires repeated exposure to mitigate pathological cardiac remodeling. We did not test the effects of transient dosing in our TAC model but speculate that withdrawal of THZ1 treatment would likely lead to reversal of protection, as we have seen with small molecule inhibitors of BET bromodomains¹⁰. Future studies will be aimed at elucidating the temporal effect of THZ1 treatment on gene expression and cardiac function.

Authors could carry out ChIP-seq experiments for Ser7p and calculate the pol II pausing index (the ratio of pol II density at the promoter/gene body), which should be modified with CDK7 inhibition.

We agree that assessment of chromatin localization of Pol II Ser7P in NRVM could add additional mechanistic insight. Despite extensive efforts, we have been technically unable to generate interpretable ChIP data for Ser2P, 5P or 7P in primary cardiomyocytes. We note that the total Pol II ChIP-seq data presented in the manuscript have been very technically challenging to generate. Major reasons for this technical limitation are the high structural/sarcomeric protein composition of primary cardiomyocytes (which makes efficient chromatin isolation challenging) and the culture conditions that are necessary to generate stimulated hypertrophic responses in these cells. Specifically, our total Pol II ChIP-seq required plating NRVM at sparse densities, prolonged serum starvation, and pooling of several large culture plates to achieve 5-10 million cells per ChIP -- manipulations which are essential for robust hypertrophic responses to pharmacologic agonists but significantly diminish the signal in

ChIP-seq. Furthermore, the robust reduction of Ser7P abundance with THZ1 would render a Pol II Ser7P ChIP-seq experiment, if technically feasible, extremely challenging to normalize and interpret. Our total Pol II ChIP-seq data was included to provide orthogonal corroboration of the on-target effects of THZ1 on Pol II function in cardiomyocytes. While the Pol II traveling ratio is a validated measure of elongating Pol II, we also recognize the limitations of interpreting total Pol II ChIP-seq data and we have tempered our conclusions about specific effects on Pol II elongation vs. initiation in the revised manuscript.

“We note that Pol II ChIP-seq in primary cultured neonatal cardiomyocytes, when performed under conditions that allow for hypertrophic responses, did not provide signal robustness that would allow us to make reliable correlations between locus-specific Pol II enrichment and differential gene expression by RNA-seq. Furthermore, we cannot exclude a primary and rapid effect of THZ1 on Pol II initiation, which can influence subsequent rates of Pol II elongation and is an event in the transcription cycle that likely occurs at much earlier timepoints after agonist stimulation compared to the 48-hour time point used in this experiment.”

Reviewer #2 (Remarks to the Author):

This submission explores whether inhibition of Cyclin-Dependent Kinase 7 (CDK7), the catalytic part of the trimeric CAK complex, can attenuate the pathological effects of stress-induced transcription and hypertrophy during the development of HF. It is an essential component of the transcription factor TFIIF, that is involved in transcription initiation and DNA repair. The manuscript is highly novel in that it demonstrates a role for CDK7 in initiation of maladaptive transcriptional initiation, and that pharmacological targeting upstream of genes such as Nppa and Nppb may be a potent method of attenuating progression to HFREF. A major strength of this submission is that the oncological data supporting THZ1 as a potent regulator of pathological gene transcription via CDK7 inhibition are already well-established, with safety and efficacy studies well-documented in cancer models. This submission presents a repurposing of a cancer drug for HF, which would expedite testing in this patient population. Overall, the manuscript is of high to exceptional quality, with a broad readership due and high clinical relevance. All experiments performed were appropriate with proper statistical analysis and controls. Data are compelling and well-organized to support the premise. Some questions from the reviewer as well as suggestions to improve readability and completeness are detailed below.

We thank reviewer #2 for the thorough evaluation of our study, the positive comments about the significance and quality of the work, and for the constructive comments below.

Reviewer comments:

The authors state that “Like THZ1, YKL-1-116 inhibited cellular hypertrophy and regulation of Nppa/Nppb induction at concentrations similar to those that inhibit cancer cell states.” But in comparing the IC50s of these two compounds (Fig.1F to Supp.Fig. 2B), THZ1 was tested on NRVM at doses of 1-10 nM but YKL-1-116 at 0.1-5.0 uM (100-5,000 nM) – a huge difference in functional IC50s. Which dose range is more comparable to oncologically therapeutic doses? Reviewer assumes both cannot be correct. The authors cannot make a claim that these two compounds are equivalent/comparable and should instead temper their claim to state that both are effective. However, the use of two structurally distinct pharmacological inhibitors to illustrate the importance of CDK7 and support the RNAi data in abrogating PE-induced hypertrophy was

an appropriate and well-utilized approach. Suppl.Fig.1B shows low-mid nM doses for various cell lines

We appreciate the reviewer's support for our utilization of two structurally distinct pharmacologic inhibitors and RNAi as orthogonal approaches to support the central thesis of our study (which proposes that inhibition of CDK7/12/13-containing transcriptional complexes can attenuate pathological cardiac remodeling). Of the two compounds, THZ1 has been more extensively tested in preclinical oncology studies and there are more data corroborating its low nanomolar potency (Supplemental Figure 2A). While the YKL-1-116 tool compound was developed to have increased selectivity for CDK7, we recognize that it is not as potent as THZ1 in our cardiomyocyte assay and requires ~10-100 fold higher concentration to inhibit cardiomyocyte hypertrophy in vitro, a concentration range which may have some polypharmacy in this context. As suggested by the reviewer, we have tempered our conclusions pertaining to the isoform selectivity of the YKL-1-116 compound and instead highlight that the similar phenotypic effect of YKL-1-116 and THZ1 provide orthogonal support for our hypothesis using two structurally distinct chemical probes. In addition, we now recognize that the full therapeutic effects of THZ1 may indeed be mediated by its ability to inhibit three CDKs (7, 12 and 13), a conclusion that we support with new siRNA experiments in which we perform triple-knockdown of CDK7/12/13 (Figure 1; see response to reviewer 1). Based on this new data, we have reframed the revised manuscript to highlight that the combined activity of CDK7/12/13 drives the hypertrophic response.

In Fig.2B, authors should at least briefly detail any common pathways or functions for the genes in Groups 1 and 3 which were altered at baseline with THZ1 treatment. Of note, hsp90, a critical cardiac chaperone protein, is listed. These gene clusters may represent important basal effects on cardiomyocytes which are not assessable via hypertrophic measurement (e.g., contractility, Ca²⁺ flux, UPR).

We thank the reviewer for this suggestion and have performed gene ontology analysis for each gene cluster. Both clusters 1 and 3 and enriched for genes that represent metabolic processes. Cluster 1 top GO terms include "cellular metabolic process, nitrogen compound metabolic process, and metabolic process". Cluster 3 top GO terms include "nucleobase-containing compound metabolic process, nucleic acid metabolic process, and heterocycle metabolic process". These common pathways may reflect a role for CDK7/12/13 in regulating metabolic plasticity in cardiac myocytes. These data are included in Supplemental Table 4 of the revised manuscript.

TAC (60d +/- CDK7i) as a model of HFrEF is appropriate and the rescue effects are marked. In Supp.Fig.3G, why was Pol II Ser7P only measured? Were there no changes to Pol II Ser2P/Ser5P in the TAC mice without pharmacological inhibition?

In the revised manuscript, we have now added western blot data for Ser2P and Ser5P abundance in heart from the TAC model (right panel; included as Supplemental Figure 4G in revised manuscript). Similar to what we have observed for Ser7P, we find that bulk Ser2P abundance is increased with TAC and attenuated by THZ1. We did not observe significant changes in bulk Pol II Ser5P abundance, a signal that does not appear to be dynamically responsive to TAC or THZ1 treatment, possibly due to compensation by other

kinases or a limited ability for bulk tissue protein lysates to detect changes at this specific phosphorylation site. These findings parallel our observations using THZ1 in cultured cardiomyocytes and support that THZ1 is having an on-target pharmacodynamic effect in the adult heart *in vivo* in our TAC model.

Authors assessed overall body weight of TAC and sham mice treated with THZ1, but were cachexia or cell death in skeletal or SMC myocytes measured? The question arises whether long-term clinical administration of THZ1 would have detrimental effects on other cell types (particularly those with high turnover) due to its transcriptional inhibitory effects. The authors state that at the doses used, “THZ1... has been validated to provide target coverage and efficacy in murine tumor xenograft studies and is well tolerated in mice,” but do not mention whether patients on THZ1 therapy for tumor subtypes experience any side effects. Of note, CDK7 is important for neuronal plasticity. Reviewer suggests that some discussion about the feasibility of administration to HFrEF patients be included, particularly any potential drawbacks.

We strongly agree with the reviewer that systemic exposure to compounds like THZ1 can have on-target liabilities in extracardiac tissues. The aim of our study was to provide initial proof-of-concept that CDK7/12/13 inhibition with THZ1, at doses studied in murine models of cancer, could blunt maladaptive transcriptional activation in the heart *in vitro* and *in vivo*. While we found that mice exposed to THZ1 at the doses used in this study did not die, lose body weight or drop their blood pressure, we did not perform extensive toxicologic analyses *in vivo*. Prior reports of THZ1 administration in murine tumor xenograft models at the same doses used in our TAC study also reported no changes in animal body weight or behavior⁹. We also note that acute systemic deletion of *Cdk7* in weaning-age mice (using a tamoxifen-inducible Cre driven by the Ubiquitin-C promoter) was surprisingly well-tolerated into adulthood²⁰, suggesting that the therapeutic index of pharmacological CDK7 inhibition might be tolerable in certain contexts. We agree with the reviewer that on-target liabilities of systemic CDK7 inhibition (e.g. effects on neuronal plasticity or on highly proliferative tissues) are still likely and remain a very important consideration, particularly for an indication such as heart failure. We acknowledge this very important issue in the Discussion section of the revised manuscript in a very objective and measured manner.

“Our finding that THZ1 can also suppress pathologic remodeling in human iPSC-CMs at low nanomolar concentration supports the concept that manipulating stress-mediated transcriptional signaling might be an approach to limit adverse cardiac remodeling during human HFrEF pathogenesis. However, we acknowledge that systemic exposure to molecules such as THZ1 may have on-target toxicity in extracardiac organs, including effects on neuronal plasticity²¹ and possibly in highly proliferative tissue compartments such as the intestinal epithelium or bone marrow. Given the generally high bar for safety for chronically dosed cardiovascular therapeutics, future studies detailing the on-target liabilities of CDK7/12/13 inhibition and the precise cell compartments mediating therapeutic efficacy will be required to consider such a strategy in the treatment of heart failure. In contrast to several cancer drugs that cause cardiotoxicity, our data suggest that molecules like THZ1 may be a privileged class of anticancer therapeutics that have cardioprotective properties. More broadly, this work supports the contention that HFrEF pathogenesis, like cancer, features a general dependency on transcription that might be therapeutically exploited.”

We hope that our initial proof-of-concept study using THZ1 in heart failure prompts further research into the role of CDK7/12/13 in the failing heart and in extracardiac tissues and catalyzes testing of next generation chemical probes in models of heart disease.

Reviewer #3 (Remarks to the Author):

In this study, the authors investigate a role for the transcriptional kinase Cdk7 in pathogenesis of heart failure with reduced ejection fraction (HFrEF), and demonstrate potential therapeutic efficacy of inhibitors targeting this kinase in preclinical models. They show in a rat cardiomyocyte culture model that depletion of Cdk7 by RNAi can limit hypertrophy and attenuate pathological gene expression changes induced by the alpha1-adrenergic agonist phenylephrine (PE), and replicate these findings with the covalent inhibitor THZ1, which has moderate selectivity for Cdk7. They validate these effects in vivo in a mouse model of heart failure, and in human iPSC-derived cardiomyocytes. There are some mechanistic questions I would like to see addressed, and some methodological issues I would hope to see clarified, before recommending publication, but overall this is a well-executed study, and the results have potential therapeutic applications to a major human health problem. My specific concerns and questions:

1. In Fig. 1A, the si-Cdk7 cells appear larger than si-cntrl in the absence of PE. Is this difference significant (statistically) and reproducible? If so, can the authors comment on what it might signify?

The trend towards a difference in cell size observed in cultured cardiomyocytes with si-Cdk7 is reproducible, but not statistically significant (Supplemental Figure 1A). As described in our methods, maintaining these cells in a relatively quiescent state that permits robust agonist-stimulated responses requires plating at low density and serum starvation. We speculate that this small trend for a baseline effect may be driven by a mild increase in cellular stress under these culture conditions that is potentiated via the particular Cdk7 siRNA that we used in this experiment. Importantly, this non-significant trend towards increased cell area was not accompanied by increased expression of the classic stress markers *Nppa* and *Nppb*. Furthermore, we have found that administration of THZ1 to adult mice in vivo does not decrease heart weight or cardiomyocyte size in the control (sham surgery) group (Figure 4), supporting that THZ1 does not have a significant effect on the baseline state in mature, adult cardiomyocytes in the context of the intact organism.

We highlight that we have generated new data (see panel above; included in Figure 1 of the revised manuscript) demonstrating the effect of triple knockdown of Cdk7, 12, and 13 on cell size. Notably, we do not observe the same trend in basal/unstimulated cell size during triple

knockdown (Figure 1F). Similarly, we do not see this baseline effect using 10nM THZ1 (see below; Figure 1B).

2. The effects of Cdk7 knockdown on Pol II CTD phosphorylation in PE-treated cardiomyocytes are quite modest (Fig. 1C) and should be quantified. It should also be pointed out that CTD phosphorylation changes due to Cdk7 knockdown (or inhibition) are likely a mix of direct and indirect effects (i.e., reduced transcription leading to decreased overall Pol II phosphorylation); Cdk7 is not thought to phosphorylate Ser2, and as the authors point out, multiple kinases can modify Ser5 and Ser7 *in vivo*.

We agree with the reviewer that the effects of isolated CDK7 knockdown on Pol II CTD phosphorylation shown in the initial manuscript were modest, and do not adequately or consistently recapitulate the potent pharmacology of THZ1 (a chemical probe which can inhibit CDK7, 12 and 13). Furthermore, we agree that this dataset and its interpretation can be substantially improved. Similar comments were also made by Reviewer #1. We have repeated the western blots for Pol II phosphoforms after CDK7 knockdown NRVM in an independent experiment (Supplemental Figure 1). We reproducibly find that CDK7 knockdown results in attenuation of Pol II phosphorylation. We have included densitometry to quantify these changes in Pol II CTD phosphorylation (see below; provided as Figure 1G and Supplemental Figures 1E-G of revised manuscript).

A mounting body of evidence now supports that CDK12 and CDK13 can indeed compensate for CDK7, particularly when the latter is inhibited^{7,9}. This functional redundancy is corroborated by our observation that siRNA-mediated knockdown of CDK7 alone does not fully approximate all aspects of the potent anti-hypertrophic effects of THZ1, suggesting that CDK12 and CDK13 may indeed be compensating. This is also supported by our observation that the small molecule probe THZ1, which can simultaneously inhibit CDK7, 12 and 13⁹, blocks cardiomyocyte hypertrophy and RNAPII phosphorylation much more consistently and robustly than does siRNA-mediated knockdown of Cdk7.

In the revised manuscript, we no longer focus exclusively on CDK7 inhibition and have completely reframed that narrative around the combined effects of CDK7/12/13 inhibition, which more accurately reflects the pharmacology of the small molecule THZ1. We have added an extensive series of experiments in which we perform individual knockdown of Cdk7, Cdk12, and Cdk13 in NRVM to illustrate that each manipulation alone does not recapitulate the full potency of THZ1, while triple knockdown of Cdk7/12/13 (which better approximates the pharmacological effect of THZ1) has robust effects on cardiomyocyte hypertrophy and Pol II phosphorylation. The triple knockdown data is highlighted below for Reviewer 3 (the data on individual

knockdown of Cdk7, Cdk12 or Cdk13 is provided in Supplementary Figure 1 of the revised manuscript).

3. On a related topic, the most dramatic effect of THZ1 in this system is on Ser2 phosphorylation, which might reflect the indirect mechanism invoked in point 2 (and by the authors later in the text) but could also be due to the significant potency of THZ1 towards Cdk12 and Cdk13, two transcriptional kinases that have been directly implicated in Ser2 phosphorylation (and are likely responsible for some of THZ1's anti-cancer effects—see for example, ref. 36). That is probably the case for the potentially therapeutic effects described here: Note that the doses of the more Cdk7-selective inhibitor YKL-1-116 required to block cell hypertrophy or prevent induction of Nppa and Nppb are higher than those of THZ1 (cf. Supp. Fig 2B,C and Fig. 1E,F) even though YKL-1-116 is at least as potent an inhibitor of Cdk7 (see ref. 29). To define Cdk7's role in HFrEF with any precision (or the potential efficacy of Cdk7-selective inhibition in ameliorating it), more extensive experiments with more selective compounds such as YKL-1-116 would be needed. In the meantime I suggest the authors tone down or qualify some of their stronger statements about Cdk7 being the central transcriptional CDK in HFrEF pathogenesis, or the primary therapeutic target of THZ1 (e.g. in the title).

This is indeed a critical point that was also raised by Reviewer's 1 and 2. As mentioned in the response to comment #2 above, we no longer focus exclusively on CDK7 inhibition and have completely reframed our narrative around the combined effects of CDK7/12/13 inhibition, which more accurately reflects the pharmacology of the small molecule THZ1. The reviewer is indeed correct that Ser2 phosphorylation of the Pol II CTD is not a robust primary target of CDK7. The observed reduction in Ser2P abundance, which correlates with bulk quantities of elongating Pol

II, may reflect secondary effects of inhibiting upstream steps in the Pol II transcription cycle (which are reflected by the abundance of Ser5P and Ser7P). Inhibition of these upstream steps in transcription can ultimately reduce the abundance of elongating Pol II¹². In addition, CDK12, which is also a key target of THZ1, has been previously demonstrated to phosphorylate Pol II Ser2P both in vitro and in vivo^{13,14}, as noted by this Reviewer. We have included new text in the manuscript to highlight the interdependence of Pol II CTD phosphorylation sites and the additional effects of THZ1 on CDK12 and 13.

“THZ1 also inhibits two related Pol II CTD kinases, CDK12 and CDK13, by covalently binding accessible cysteine residues that are homologous to C312 of CDK7”

“THZ1 attenuated the PE-dependent increase in Ser2P and Ser7P (Figure 1C). We did not observe a significant effect on bulk abundance of Ser5P in cardiomyocytes, which may reflect redundant regulation of this phosphoform by other Pol II CTD kinases¹⁵⁻¹⁸.”

“Furthermore, we cannot exclude a primary and rapid effect of THZ1 on Pol II initiation, which can influence subsequent rates of Pol II elongation and is an event in the transcription cycle that likely occurs at much earlier timepoints after agonist stimulation compared to the 48-hour time point used in this experiment.”

“To validate for our findings using THZ1, we also assessed the anti-hypertrophic effect of the small molecule probe YKL-1-116¹⁹ (Supplemental Figure 2A), which has substantially lower potency against CDK12/13. YKL-1-116 inhibited cellular hypertrophy and Nppa/Nppb induction, but required concentrations 50-100 fold higher than THZ1. These data suggest that potent and combined inhibition of CDK7/12/13 is required for maximal suppression of the hypertrophic stress response in cardiomyocytes.”

As suggested by the reviewer, we have also tempered all the language about CDK7 being the central transcriptional CDK in HFrEF pathogenesis or the exclusive therapeutic target of THZ1. We have also changed the title of the manuscript to reflect that the effects we are observing on pathological cardiac remodeling are due to inhibition of multiple transcriptional kinases (i.e. CDK7/12/13).

The new title for the revised manuscript is: *“Targeting Transcription in Heart Failure via CDK7/12/13 inhibition”*

4. A minor point, but I would recommend the authors provide a brief description in the main text of how they calculated “traveling ratio” (Fig. 2E). Unfortunately, this term has been used in the past to refer to either the ratio of Pol II over the TSS versus the gene body (which has also been called the “pause index”), or its reciprocal ratio (i.e., gene body over TSS). Although the former definition (TR=PI) seems to have won out, and appears to be the one used here, it would help to say so explicitly to avoid ambiguity.

We thank the reviewer for this comment and agree that explicit clarification of “traveling ratio” is important. We have included the following text for clarification:

“Finally, we assessed the Pol II traveling ratio (defined here as the ratio of Pol II occupancy at the TSS over the gene body), a widely used indicator of genome-scale transcription elongation, for genes regulated by THZ1 during PE-mediated stress.”

5. This result is somewhat surprising and potentially interesting, but requires some context and additional explanation. Previous analyses of Pol II occupancy after Cdk7 inhibition (or THZ1 treatment) indicated decreased density across gene bodies with, if anything, greater depletion over the TSS (see for example ref. 3). That pattern predicts an unchanged or decreased PI/TR (see point 4) and is consistent with the role of Cdk7 activity in recruiting the pausing factors DSIF and NELF to establish the promoter-proximal pause (Glover-Cutter et al., Mol. Cell Biol. 29:5455, 2009; Laroche et al., Nat. Struct. Mol. Biol. 19:1108, 2012). Can the authors comment on whether this behavior is specific to the genes induced by PE (i.e., does the TR go up or down in the cluster 1 genes repressed by THZ1 treatment in the absence of PE)? Another potential source of the difference with previous studies is the time of treatment with THZ1, but I could not find that information for the RNA-seq or CHIP-seq experiments. Was it the 48-hr time of treatment used in Fig. 1E? (If so, that could explain the discrepancy since the previous studies used much shorter durations of inhibition.)

We used a 48-hour time point for these total Pol II ChIP-seq studies in cultured cardiomyocytes. The effect on TSS vs. gene body enrichment for Pol II that we observed may indeed be due to our assessment of a relatively late timepoint after exposure to hypertrophic stimulation. It remains possible that ChIP-seq performed at very early timepoints (e.g. minutes) after THZ1 exposure might have shown depletion of Pol II enrichment at the TSS.

We note that the total Pol II ChIP-seq data presented in the manuscript have been very technically challenging to generate. Major reasons for this technical limitation are the high structural/sarcomeric protein composition of primary cardiomyocytes (which makes efficient chromatin isolation challenging) and the culture conditions that are necessary to generate stimulated hypertrophic responses in these cells. Specifically, our total Pol II ChIP-seq required plating NRVM at sparse densities, subjecting the cells to prolonged serum starvation, and pooling of several large culture plates to achieve 5-10 million cells per ChIP -- manipulations which are essential for eliciting robust hypertrophic responses to pharmacologic agonists but significantly diminish the signal in ChIP-seq. As such, we currently lack the signal resolution to reliably compare Pol II enrichment at genes that are induced vs. repressed/unaffected by PE in a rigorous manner. Dissecting Pol II dynamics in a more incisive manner will likely require future experiments that assess nascent transcription on a genome-wide basis, such as PRO-seq. PRO-seq remains extremely technically challenging to perform in primary cultured cardiomyocytes in the context of hypertrophic stimulation and has never been reported in this experimental system. We are actively trying to develop these techniques in our laboratory for future studies. Our total Pol II ChIP-seq data was principally meant to support the western blot data and provide some orthogonal corroboration of the on-target effects of THZ1 on Pol II in primary cultured cardiomyocytes. While the Pol II traveling ratio is a reasonable surrogate for elongating Pol II, we also recognize its limitations and we have tempered our conclusions about specific effects on Pol II elongation vs. initiation in the revised manuscript.

6. In the PCA (Fig. 4B), is there any explanation for the large shift along PC2 in 2 of the 4 mice that received TAC + THZ1?

The left ventricular tissue samples that were processed for RNA-seq were a 2mm short axis slice at the mid-ventricular level (i.e. a "donut slice"), taken at a late time point in this model (8 weeks after the TAC surgery). Samples were selected based on whether the animals were representative of the cardiac phenotype of the particular experimental group (in this case, they most closely represented the mean heart weight and ejection fraction of their respective experimental group). The RNA samples and library preps all passed standard quality control metrics. One explanation for this variability is due to the patchy nature of adverse cardiac

remodeling, inflammation, fibrosis, and subendocardial injury that can occur after prolonged pressure overload in the TAC model – features which can lead to biological and spatial heterogeneity in a particular tissue sample, especially after 8 weeks of pressure overload. Differences in an animal's plasma concentrations of THZ1 or pharmacodynamic response to THZ1 at the time of harvest may also contribute to these effects, which may also become more varied in the later stages of heart failure. Despite this variability in the TAC-THZ1 group that is observed in this two-dimensional PCA plot, we were able to apply rigorous statistical criteria to determine differentially expressed genes with high confidence.

References cited in the Reviewer Response Document

- 1 Sano, M. *et al.* Activation and function of cyclin T-Cdk9 (positive transcription elongation factor-b) in cardiac muscle-cell hypertrophy. *Nature medicine* **8**, 1310-1317, doi:10.1038/nm778 (2002).
- 2 Yankulov, K., Yamashita, K., Roy, R., Egly, J. M. & Bentley, D. L. The transcriptional elongation inhibitor 5,6-dichloro-1-beta-D-ribofuranosylbenzimidazole inhibits transcription factor IIH-associated protein kinase. *J Biol Chem* **270**, 23922-23925, doi:10.1074/jbc.270.41.23922 (1995).
- 3 Rickert, P., Corden, J. L. & Lees, E. Cyclin C/CDK8 and cyclin H/CDK7/p36 are biochemically distinct CTD kinases. *Oncogene* **18**, 1093-1102, doi:10.1038/sj.onc.1202399 (1999).
- 4 Schang, L. M. Cyclin-dependent kinases as cellular targets for antiviral drugs. *J Antimicrob Chemother* **50**, 779-792, doi:10.1093/jac/dkf227 (2002).
- 5 Sehgal, P. B., Darnell, J. E., Jr. & Tamm, I. The inhibition by DRB (5,6-dichloro-1-beta-D-ribofuranosylbenzimidazole) of hnRNA and mRNA production in HeLa cells. *Cell* **9**, 473-480, doi:10.1016/0092-8674(76)90092-1 (1976).
- 6 Eom, G. H. *et al.* Casein kinase-2alpha1 induces hypertrophic response by phosphorylation of histone deacetylase 2 S394 and its activation in the heart. *Circulation* **123**, 2392-2403, doi:10.1161/CIRCULATIONAHA.110.003665 (2011).
- 7 Zeng, M. *et al.* Targeting MYC dependency in ovarian cancer through inhibition of CDK7 and CDK12/13. *Elife* **7**, doi:10.7554/eLife.39030 (2018).
- 8 Olson, C. M. *et al.* Development of a Selective CDK7 Covalent Inhibitor Reveals Predominant Cell-Cycle Phenotype. *Cell Chem Biol* **26**, 792-803 e710, doi:10.1016/j.chembiol.2019.02.012 (2019).
- 9 Kwiatkowski, N. *et al.* Targeting transcription regulation in cancer with a covalent CDK7 inhibitor. *Nature*, doi:10.1038/nature13393 (2014).
- 10 Alexanian, M. *et al.* A transcriptional switch governs fibroblast activation in heart disease. *Nature* **595**, 438-443, doi:10.1038/s41586-021-03674-1 (2021).
- 11 Tabula Muris, C. *et al.* Single-cell transcriptomics of 20 mouse organs creates a Tabula Muris. *Nature* **562**, 367-372, doi:10.1038/s41586-018-0590-4 (2018).
- 12 Nilson, K. A. *et al.* THZ1 Reveals Roles for Cdk7 in Co-transcriptional Capping and Pausing. *Mol Cell* **59**, 576-587, doi:10.1016/j.molcel.2015.06.032 (2015).
- 13 Blazek, D. *et al.* The Cyclin K/Cdk12 complex maintains genomic stability via regulation of expression of DNA damage response genes. *Genes Dev* **25**, 2158-2172, doi:10.1101/gad.16962311 (2011).
- 14 Bartkowiak, B. *et al.* CDK12 is a transcription elongation-associated CTD kinase, the metazoan ortholog of yeast Ctk1. *Genes Dev* **24**, 2303-2316, doi:10.1101/gad.1968210 (2010).
- 15 Liao, S. M. *et al.* A kinase-cyclin pair in the RNA polymerase II holoenzyme. *Nature* **374**, 193-196, doi:10.1038/374193a0 (1995).

- 16 Sun, X. *et al.* NAT, a human complex containing Srb polypeptides that functions as a negative regulator of activated transcription. *Mol Cell* **2**, 213-222 (1998).
- 17 Trigon, S. *et al.* Characterization of the residues phosphorylated in vitro by different C-terminal domain kinases. *J Biol Chem* **273**, 6769-6775 (1998).
- 18 Czudnochowski, N., Bosken, C. A. & Geyer, M. Serine-7 but not serine-5 phosphorylation primes RNA polymerase II CTD for P-TEFb recognition. *Nat Commun* **3**, 842, doi:10.1038/ncomms1846 (2012).
- 19 Kalan, S. *et al.* Activation of the p53 Transcriptional Program Sensitizes Cancer Cells to Cdk7 Inhibitors. *Cell Rep* **21**, 467-481, doi:10.1016/j.celrep.2017.09.056 (2017).
- 20 Ganuza, M. *et al.* Genetic inactivation of Cdk7 leads to cell cycle arrest and induces premature aging due to adult stem cell exhaustion. *EMBO J* **31**, 2498-2510, doi:10.1038/emboj.2012.94 (2012).
- 21 He, G. *et al.* Cdk7 Is Required for Activity-Dependent Neuronal Gene Expression, Long-Lasting Synaptic Plasticity and Long-Term Memory. *Front Mol Neurosci* **10**, 365, doi:10.3389/fnmol.2017.00365 (2017).

REVIEWERS' COMMENTS

Reviewer #1 (Remarks to the Author):

No further questions.

Reviewer #2 (Remarks to the Author):

The reviewer thanks the authors for their careful and detailed responses to the reviewer's questions and comments, and feels that the additions and changes included in the re-submission of this manuscript significantly improve its quality and suitability for publication.

Reviewer #3 (Remarks to the Author):

This is a revised version of a manuscript I reviewed previously, now reframed as an investigation of the role of multiple transcriptional CDKs in heart failure. The authors have addressed my concerns adequately. I also believe they have dealt sufficiently with the valid concerns of other reviewers. Although responding to the reviews required a more nuanced topline conclusion and mechanistic interpretations, the study remains of high interest and importance. There are a few places where I disagree with some of the new interpretations and explanations, and also some issues of presentation that distract or divert from the flow of the paper, but these should be easy to fix. My specific concerns:

1. Line 250 and elsewhere: I disagree with the statement that the data support "overlapping roles" of CDK7, 12 and 13. From a mechanistic standpoint it is difficult to see how this could be true: CDK7 performs its transcriptional function as a subcomplex within TFIIH and there is no evidence that any other CDK can replace or supplant it in the holoTFIIH complex or even compete for the same cyclin (which happens in other contexts, e.g. when the CDK2 gene is deleted and CDK1 takes over its cell-cycle functions). The more cautious (and likely) explanation is that multi-CDK inhibition or depletion (neither of which is likely to abolish activity completely) is having additive or synergistic effects by impairing multiple, non-overlapping functions in the transcription cycle. I have the same objection to the use of "compensation" to refer to the same phenomenon in the rebuttal letter (which I don't think appears in the manuscript).
2. Lines 257-262: I agree with this rebuttal to Reviewer 1 in that multiple, conditional knockouts of CDK7, -12 and -13 to try to recapitulate the effects seen here with THZ1 and RNAi would be a huge and risky undertaking, and definitely beyond the scope of this study. In my opinion, the model proposed here is strongly supported by the data presented. For future studies (also beyond the scope of this report), I wonder if the authors have considered using THZ1-resistant alleles of the three suspected targets (i.e. C312S mutation in CDK7 and equivalent cysteine substitutions in CDK12 and -13). In cancer cells, CDK7-C312S provided substantial resistance to effects of THZ1 (ref. 3), indicating that CDK7 inhibition was necessary, although as it turned out not sufficient, for its cell-killing effects.
3. A relatively minor point on line 94: The text describes "a sensitivity that is significantly lower..." when the sensitivities in this system to THZ1 are actually higher than those in published cancer models, because the IC50s are lower (or roughly equal).
4. A minor point: Many of the figure call-outs are out of order, especially with respect to supplemental figures. For example, Supplemental Figure 1 is first cited in line 110, after both Supplemental Figures 2 and 3. (There are several other examples.)

5. Line 268: A minor point, but the correct term is "P-TEFb" not "PTEF-b."
6. Line 269: I believe the authors mean the benzimidazole derivative DRB (5,6-dichloro-1- β -D-ribofuranosylbenzimidazole) when they say "benzimidazole" but they are not the same.

Robert P. Fisher

Reviewer #3 (Remarks to the Author):

This is a revised version of a manuscript I reviewed previously, now reframed as an investigation of the role of multiple transcriptional CDKs in heart failure. The authors have addressed my concerns adequately. I also believe they have dealt sufficiently with the valid concerns of other reviewers. Although responding to the reviews required a more nuanced topline conclusion and mechanistic interpretations, the study remains of high interest and importance. There are a few places where I disagree with some of the new interpretations and explanations, and also some issues of presentation that distract or divert from the flow of the paper, but these should be easy to fix. My specific concerns:

We thank reviewer #3 for their careful review of the revised manuscript, for acknowledging the overall significance of the work, and taking the time to provide highly constructive comments. We agree that addressing these issues will improve the manuscript.

1. Line 250 and elsewhere: I disagree with the statement that the data support “overlapping roles” of CDK7, 12 and 13. From a mechanistic standpoint it is difficult to see how this could be true: CDK7 performs its transcriptional function as a subcomplex within TFIIH and there is no evidence that any other CDK can replace or supplant it in the holoTFIIH complex or even compete for the same cyclin (which happens in other contexts, e.g. when the CDK2 gene is deleted and CDK1 takes over its cell-cycle functions). The more cautious (and likely) explanation is that multi-CDK inhibition or depletion (neither of which is likely to abolish activity completely) is having additive or synergistic effects by impairing multiple, non-overlapping functions in the transcription cycle. I have the same objection to the use of “compensation” to refer to the same phenomenon in the rebuttal letter (which I don’t think appears in the manuscript).

This is a very important point and we thank the reviewer for helping us convey the mechanistic interpretations of our data in a more accurate manner. We acknowledge the reviewer’s point that no existing evidence to support a compensatory role for CDK12 or CDK13 in the holoTFIIH complex upon CDK7 inhibition or depletion. We have therefore removed any discussion of compensation by other CDKs and reframed the text to more appropriately reflect the supported conclusion – i.e., that CDK7, 12 and 13 have non-overlapping functions in the transcription cycle. We agree that multi-CDK inhibition and/or depletion is having additive or synergistic effects on cardiac stress responses by impairing multiple, non-overlapping functions in the transcription cycle. Given the clarity and accuracy of Reviewer 3’s suggested interpretation above, we have used very similar language in the manuscript text.

“Our in vitro data using chemical inhibitors and siRNA are consistent with a mechanism where CDK7, 12 and 13 have non-overlapping functions in the transcription cycle. In this manner, multi-CDK inhibition of CDK7, 12 and 13 using THZ1 or siRNA is likely exerting additive or synergistic effects on cardiac stress responses by impairing their multiple, non-overlapping functions in the transcription cycle, supporting the concept that inhibition of all three kinases is required to consistently and potently suppress adverse cardiac remodeling.”

2. Lines 257-262: I agree with this rebuttal to Reviewer 1 in that multiple, conditional knockouts of CDK7, -12 and -13 to try to recapitulate the effects seen here with THZ1 and RNAi would be a huge and risky undertaking, and definitely beyond the scope of this study. In my opinion, the model proposed here is strongly supported by the data presented. For future studies (also beyond the scope of this report), I wonder if the authors have considered using THZ1-resistant

alleles of the three suspected targets (i.e. C312S mutation in CDK7 and equivalent cysteine substitutions in CDK12 and -13). In cancer cells, CDK7-C312S provided substantial resistance to effects of THZ1 (ref. 3), indicating that CDK7 inhibition was necessary, although as it turned out not sufficient, for its cell-killing effects.

We have contemplated performing such cysteine-substitution studies in the context of cardiac stress responses. However, our studies require manipulation of primary cells, as there are no cardiomyocyte cell lines (or lines of other relevant cardiac cell types) that adequately model heart failure-related stress responses. Therefore, generating the cysteine-substitution mutant cells to study in conjunction with cysteine-targeted covalent CDK inhibitors would require generation of gene-targeted rodents that harbor such coding variants in the germline, followed by extraction of primary cells from the hearts of these animals. An alternative approach would be to engineer gene-targeted human iPSCs harboring the relevant cysteine-mutations, differentiate them into cardiomyocytes, and study their responses to stress in the context of chemical probes like THZ1. We agree with Reviewer 3 that future studies utilizing such THZ1-resistant alleles would indeed be quite interesting to pursue and could further illuminate how these covalent CDK inhibitors alter transcription in heart failure.

We have added the following sentence to the Discussion section to highlight the utility of cysteine substitution variants of CDK7/12/13:

“Future studies using cardiovascular cells harboring a CDK7 cysteine substitution allele (Cys312Ser) that is resistant to covalent chemical inhibitors like THZ1 (or equivalent cysteine substitution alleles for CDK12/13), will also help dissect the non-overlapping functions of these three kinases in cardiac stress responses.”

3. A relatively minor point on line 94: The text describes “a sensitivity that is significantly lower...” when the sensitivities in this system to THZ1 are actually higher than those in published cancer models, because the IC50s are lower (or roughly equal).

We thank Reviewer 3 for picking this up. We intended to say that the cardiomyocytes were more sensitive to the effects of THZ1 because the IC50s in these assays were lower concentrations than have been observed in studies of growth inhibition in several cancer cell lines. We have corrected this sentence in the text to reflect the higher sensitivity of THZ1 observed in our experimental system.

“The IC50 of THZ1 in these cardiomyocyte assays was 5-10nM, which reflects a sensitivity to THZ1 that is higher than what has been observed in studies of growth inhibition of several cancer cell types.”

4. A minor point: Many of the figure call-outs are out of order, especially with respect to supplemental figures. For example, Supplemental Figure 1 is first cited in line 110, after both Supplemental Figures 2 and 3. (There are several other examples.)

We have corrected the ordering of the main figure and supplemental figure callouts throughout the manuscript.

5. Line 268: A minor point, but the correct term is “P-TEFb” not “PTEF-b.”

We have made this correction.

6. Line 269: I believe the authors mean the benzimidazole derivative DRB (5,6-dichloro-1- β -D-ribofuranosylbenzimidazole) when they say “benzimidazole” but they are not the same.

We have made this correction to clarify our reference to 5,6-dichloro-1- β -D-ribofuranosylbenzimidazole.

“Using pleiotropic chemical inhibitors such as 5,6-dichloro-1- β -D-ribofuranosylbenzimidazole (DRB) or flavopirodol...”